# Pareto Optimization for Active Learning under Out-of-Distribution Data Scenarios

## Abstract

Pool-based Active Learning (AL) has achieved great success in minimizing labeling costs by sequentially selecting the most informative unlabeled samples from a large unlabeled data pool and querying their labels from oracle/annotators. However, existing AL sampling schemes might not work well under out-of-distribution (OOD) data scenarios, where the unlabeled data pool contains data samples that do not belong to the pre-defined categories of the target task. Achieving good AL performance under OOD data scenarios is a challenging task due to the natural conflict between AL sampling strategies and OOD sample detection – both more informative in-distribution (ID) data and OOD data in unlabeled data pool may be assigned high informativeness scores (e.g., high entropy) during AL processes. In this paper, we propose a Monte-Carlo Pareto Optimization for Active Learning (POAL) sampling scheme, which selects optimal subsets of unlabeled samples with *fixed batch size* from the unlabeled data pool. We cast the AL sampling task as a multi-objective optimization problem and utilize Pareto optimization based on two conflicting objectives: (1) the typical AL sampling scheme (e.g., maximum entropy) and (2) the confidence of not being an OOD data sample. Experimental results show the effectiveness of our POAL on classical Machine Learning (ML) and Deep Learning (DL) tasks.

## 1 Introduction

In real-life applications, huge amounts of unlabeled data are easily obtained, but labeling them would be expensive (Shen et al., 2004). AL aims to solve this problem – it achieves greater accuracy with less training data by sequentially selecting the most informative instances and then querying their labels from oracles/annotators (Zhan et al., 2021b). Current AL methods have been tested on well-studied datasets (Kothawade et al., 2021) like *MNIST* (Deng, 2012) and *CIFAR10* (Krizhevsky et al., 2009). These datasets are simple and clean. However, in realistic scenarios, when collecting unlabeled data, unrelated data (i.e., out-of-domain data) might be mixed in with the task-related data, e.g., images of letters when the task is to classify images of digits (Du et al., 2021). Most AL methods are not robust to OOD data scenarios. For instance, Karamcheti et al. (2021) has demonstrated empirically that collective outliers hurt AL performances under Visual Question Answering (VQA) tasks. Meanwhile, selecting and querying OOD samples that are invalid for the target model will waste the labeling cost (Du et al., 2021) and make the AL sampling process less effective.

There is a natural conflict between AL and OOD data detection. Most AL methods, especially uncertainty-based measures, prefer selecting data that are hardest to be classified by the current basic classifier (e.g., high entropy of predicted class probabilities). However, in AL, if a basic learner (e.g., Neural Network with softmax output) performs poorly on ID data, it is more likely to provide non-informative predicted probabilities (i.e., close to uniform probabilities) on OOD data (Vaze et al., 2021). During AL processes, the basic learner is not well-trained due to insufficient labeled data, and insufficient epochs in the case of deep AL. Therefore, the samples selected by AL may contain both high informative ID and OOD samples. For example, consider the Maximum Entropy (ENT) approach for AL, which is a classic uncertainty-based method (Lewis & Catlett, 1994; Shannon, 2001) that selects data whose predicted class probabilities have the largest entropy. Meanwhile, ENT is *also a typical OOD detection method* – high entropy of the predicted class distribution suggests that the input may be OOD (Ren et al., 2019). Fig. 1(top) shows an example on the *EX8* dataset (Ng, 2008), which further illustrates this conflict – a large percentage of data with high entropy scores are OOD, and thus ENT-based AL will likely select OOD data for labeling. Thus, additional measures are needed to detect OOD samples so that they are not selected for AL. For example, in Fig. 1(bottom),

the negative Mahalanobis distance (Lee et al., 2018) shows a certain negative correlation with entropy and thus could be used as an ID confidence score.

Although the OOD problem has been demonstrated to affect AL in real-life applications (Karamcheti et al., 2021), there are only a few studies on this topic (Kothawade et al., 2021; Du et al., 2021). SIMILAR (Submodular Information Measures Based Active Learning) (Kothawade et al., 2021) adopted the submodular conditional mutual information (SCMI) function as the acquisition function. They jointly model the similarity between unlabeled and labeled ID data sets and their dissimilarity with labeled OOD data sets. The estimation might not be accurate initially since both labeled ID and OOD data sets are insufficient. CCAL (Contrastive Coding AL) (Du et al., 2021) needs to pre-train extra self-supervised models like SimCLR (Chen et al., 2020), and also introduces hyper-parameters to trade-off between semantic and distinctive scores, whose values affect the final performance (see Section 4.3 in (Du et al., 2021)). These two factors limit the range of its application. We compared our work with SIMILAR and CCAL in detail and showed the superiority of our method in Appendices C and F.3.

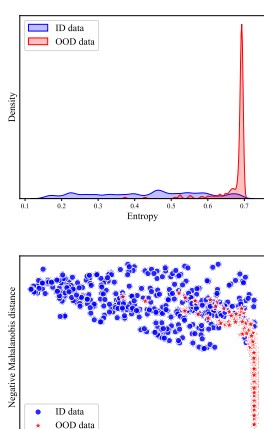

Figure 1: (top) Distribution of entropy for ID and OOD data during AL processes on *EX8*. (bottom) Scatter plot of the AL score (entropy) and ID confidence score (negative Mahalanobis distance) of unlabeled data. Larger ID score indicates data is more likely to be ID data.

In this paper, we advocate simultaneously considering the AL criterion and ID confidence when designing AL sampling strategies to address the above issues. Since the two objectives conflict, we define the AL sampling process under OOD data scenarios as a multi-objective optimization problem (Seferlis & Georgiadis, 2004). Unlike traditional methods for handling multiple-criteria based AL, such as weighed-sum optimization (Zhan et al., 2022a) or two-stage optimization (Shen et al., 2004; Zhan et al., 2022a), we propose a novel and flexible batch-mode **P**areto **O**ptimization **A**ctive **L**earning (POAL) framework. The contributions and summarization of this paper are as follows:

1. We propose AL under OOD data scenarios within a multi-objective optimization framework.
2. Our framework is flexible and can accommodate different combinations of AL and OOD detection methods according to various target tasks. In our experiments, we use ENT as the AL objective and Mahalanobis distance as ID confidence scores.
3. Naively applying Pareto optimization to AL will result in a Pareto Front with a non-fixed size, which can introduce a high computational cost. To enable efficient Pareto optimization, we propose a *Monte-Carlo (MC) Pareto optimization algorithm for fixed-size batch-mode AL.*
4. Our framework works well on both classical ML and DL tasks, and we propose pre-selecting and early-stopping techniques to reduce the computational cost on large-scale datasets.
5. Our framework has no trade-off hyper-parameter for balancing AL and OOD objectives. It is important since: i) AL is data-insufficient, there might be no validation set for tuning parameters; ii) hyper-parameter tuning in AL can be label-expensive since every change of hyper-parameter causes AL to label new data, thus provoking substantial labeling inefficiency (Ash et al., 2020).

## 2 RELATED WORK

**Pool-based Active Learning.** Pool-based AL has been well-studied in these years (Settles, 2009; Zhan et al., 2021b; Ren et al., 2021) and widely adopted in various tasks (Duong et al., 2018; Yoo & Kweon, 2019; Dor et al., 2020; Haussmann et al., 2020). Most AL methods rely on fixed heuristic sampling strategies, which follow two main branches: uncertainty- and representative/diversity-based measures (Ren et al., 2021; Zhan et al., 2022a). Uncertainty-based approaches select data that maximally reduce the uncertainty of the target basic learner (Ash et al., 2020). Typical uncertainty-based measures that perform well on classical ML tasks like Query-by-Committee (QBC) (Seung et al., 1992), Bayesian Active Learning by Disagreement (BALD) (Houlsby et al., 2011) have also been generalized to DL tasks (Wang & Shang, 2014; Gal et al., 2017; Beluch et al., 2018; Zhan et al., 2022a). Representative/diversity-based methods like $k$-Means (Zhan et al., 2022a) and Core-Set approach (Sener & Savarese, 2018) select a batch of unlabeled data most representative of the set. Uncertainty- and representative-based measures could be combined with weighted-sum or multi-stage optimization (Zhan et al., 2022a). Weighted-sum optimization combines multiple objectives

through linear combination with trade-off weights. (Yin et al., 2017) is a typical method that adopts weighted-sum optimization, selecting the most uncertain and least redundant samples as well as the most diverse. Two-stage (or multi-stage) optimization first ranks data or selects a subset based on one objective, then makes final decisions based on the other objective (Shen et al., 2004; Zhao et al., 2019; Zhan et al., 2022a). (Ash et al., 2020) computes gradient embedding of unlabeled data in the first stage (uncertainty), then clusters by KMeans++ in the second stage (diversity).

**Out-of-Distribution.** Detecting OOD data is vital in ensuring the reliability and safety of ML systems in real-life applications (Yang et al., 2021) since the OOD problem severely influences real-life decisions. For example, in medical diagnosis, the trained classifier could wrongly classify a healthy OOD sample as pathogenic (Ren et al., 2019). Existing methods compute ID/OOD confidence scores based on the predictions of (ensembles of) classifiers trained on ID data, e.g., the ID confidence can be the entropy of the predictive class distribution (Ren et al., 2019). Hendrycks & Gimpel (2017) observed that a well-trained neural network assigns higher softmax scores to ID data than OOD data, i.e., predictions for ID data have lower entropy. Follow-up work ODIN (Liang et al., 2017) amplifies the effectiveness of OOD detection with softmax score by considering temperature scaling and input pre-processing. Others (Lee et al., 2018; Cui et al., 2020) propose simple and effective methods for detecting both OOD and adversarial data samples based on Mahalanobis distance. Due to its superior performance compared with other OOD strategies (see Table 1 in (Ren et al., 2019)), our POAL framework uses Mahalanobis distance as the primary criterion for calculating the ID confidence score. Nonetheless, our framework is general, and any ID confidence score could be adopted.

**How OOD influence AL methods?** We have introduced the primary categories of AL methods, the OOD problem, and how OOD data influence AL, like uncertainty-based measures (e.g., ENT). We next discuss whether other AL methods are also affected by OOD data. As discussed in Section 1, uncertainty-based AL measures like ENT are not robust to OOD data scenarios since OOD data are naturally difficult to be classified. Representative/diversity-based methods are also not robust to OOD data since outliers/OOD samples are far from ID data and thus more likely to be selected first to cover the whole unlabeled data pool. Combined AL strategies that integrate uncertainty- and representative-based measures will also be susceptible to OOD data. Weighted-sum optimization will select data with high uncertainty and representativeness scores, which are likely to be OOD. The multi-stage optimization adopts uncertainty-based measures to select a subset and then uses representative-based measures for final selection (or vice versa). For both variants, a large proportion of OOD data will be selected in the first stage. To address these issues, we proposed our POAL, which can be adapted to various AL sampling schemes with OOD data. We conducted experiments on multiple highly-cited AL methods to validate our viewpoints, as shown in Section 4, Fig. 4.

**Multi-objective optimization.** Many real-life applications require optimizing multiple objectives that conflict with each other. For example, in the sensor placement problem, the goal is to maximize sensor coverage while minimizing deployment costs (Watson et al., 2004). Since there is no single solution that simultaneously optimizes each objective, Pareto optimization can be used to find a set of "*Pareto optimal*" solutions with optimal trade-offs of the objectives (Miettinen, 2012). A decision maker can then select a final solution based on their requirements. Compared with traditional optimization methods for multiple objectives (e.g., weighted-sum (Marler & Arora, 2010)), Pareto optimization algorithms are designed via different meta-heuristics *without any trade-off parameters* (Zhou et al., 2011; Liu et al., 2021). Besides optimization of the two conflicting objectives (AL score and ID confidence), we also need to perform batch-mode subset selection. Pareto Optimization Subset Selection (POSS) (Qian et al., 2015) solves the subset selection problem by optimizing two objectives simultaneously: maximizing the criterion function and minimizing the subset size. However, POSS does not support *more than one criterion and cannot work with fixed subset size*, thus not suitable for our task. Therefore, we propose Monte-Carlo POAL, which achieves: 1) optimization of multiple objectives; 2) no extra trade-off parameters for tuning; 3) subset selection with a fixed size.

## 3 METHODOLOGY

We introduce the problem definition of AL under OOD data scenarios, and details of our POAL.

### 3.1 PROBLEM DEFINITION AND OVERVIEW

We consider a general pool-based AL process for a $K$-class classification task with feature space $\mathcal{X}$, label space $\mathcal{Y} \in \{1, ..., K\}$ under OOD data scenarios. We assume that the oracle can provide a

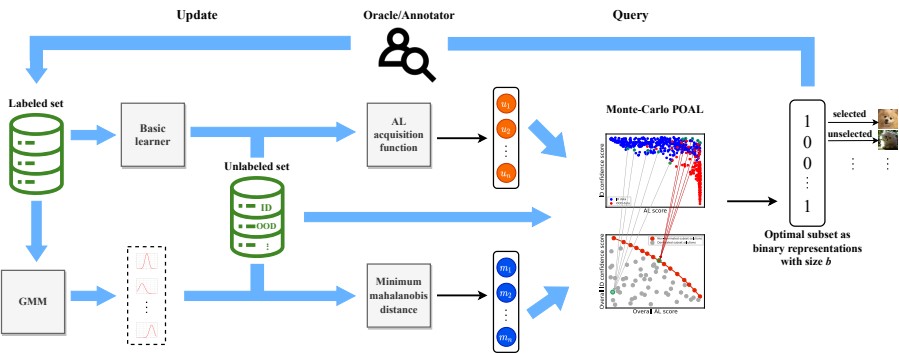

Figure 2: Overview of POAL. In each iteration, we train basic learner and estimate GMMs (in DL tasks, we estimate multivariate Gaussian distributions with low-/high- feature levels) to calculate the AL and Mahalanobis distance-based ID confidence score for unlabeled data. We then construct POAL by randomly generating candidate solutions with fixed batch size and updating the Pareto set iteratively to get the optimal solution.

fixed number of labels, denoted as budget $B$. When an OOD sample is queried, the oracle will return an "OOD" label[1] to represent data outside of the specified task. Let $B$ be the number of labels the oracle can provide, i.e., the budget. We aim to select the most informative $B$ instances with fewer OOD samples to obtain the best classification performance. To reduce the computational cost, we consider batch-mode AL, where batches of samples with fixed size $b$ are selected and queried. We denote the AL acquisition function as $\alpha(\mathbf{x}; \mathcal{A})$, where $\mathcal{A}$ refers to AL sampling strategy, and the basic learner/classifier as $f_\theta(\mathbf{x})$. We denote the current labeled set as $\mathcal{D}_l = \{(\mathbf{x}_i, y_i)\}_{i=1}^N$ and the large unlabeled data pool as $\mathcal{D}_u = \{\mathbf{x}_i\}_{i=1}^M$. The labeled data are sampled *i.i.d.* over data space $\mathcal{D}$, i.e., $\mathcal{D}_l \in \mathcal{D}$, and $N \ll M$. Under OOD data scenarios, active learners may query OOD samples whose labels are not in $\mathcal{Y}$. To simplify the problem settings, we ignore the queried OOD data since it is not useful for the classification task, and thus we only add ID samples to $\mathcal{D}_l$.

Motivated by the natural conflict between AL and OOD data detection like Fig. 1 (bottom), we propose POAL as outlined in Fig. 2. In each AL iteration, we firstly utilize the clean labeled set that only contains ID data for training a basic classifier $f_\theta(\mathbf{x})$ and constructing class-conditional Gaussian Mixture Models (GMMs) for detecting OOD samples in $\mathcal{D}_u$. Based on classifier $f_\theta(\mathbf{x})$, AL acquisition function $\alpha(\mathbf{x}; \mathcal{A})$ and the GMMs, we calculate the informativeness/uncertainty score $\mathcal{U}(\mathbf{x}_i)$ and ID confidence score $\mathcal{M}(\mathbf{x}_i)$ for each unlabeled sample $\mathbf{x}_i$ in $\mathcal{D}_u$. Let $\mathbf{s} \in \{0, 1\}^M$ be a binary vector whose element $s_i = 1$ indicates that the i-th unlabeled sample is selected, and $s_i = 0$ indicates unselected. In each AL stage, the multi-objective optimization goal is to find an optimal subset $\mathbf{s}^*$ with $b$ selected samples, simultaneously maximizes the informativeness/uncertainty and ID confidence score:

$$\mathbf{s}^* = \arg\max_{\mathbf{s} \in \{0,1\}^M} (\mathcal{U}(\mathbf{s}), \mathcal{M}(\mathbf{s})) \quad s.t. \quad \sum s_i = b, \tag{1}$$

where $\mathcal{U}(\mathbf{s}) = \sum_{i=1}^M s_i \mathcal{U}(\mathbf{x}_i)$ is shorthand for the aggregated AL score for the selected samples, and likewise for $\mathcal{M}(\mathbf{s})$. $\mathcal{U}(\cdot)$ and $\mathcal{M}(\cdot)$ will be introduced in Sections 3.2 and 3.3.

## 3.2 ACTIVE LEARNING SCORE

AL selects data by maximizing the acquisition function $a(\mathbf{x}; \mathcal{A})$, i.e., $\mathbf{x}^* = \arg\max_{\mathbf{x} \in \mathcal{D}_u} a(\mathbf{x}; \mathcal{A})$ (Gal et al., 2017). In our work, we use the AL scores for the whole unlabeled pool $\mathcal{D}_u$ for the subsequent Pareto optimization. Thus, we convert the acquisition function to a *querying density function* via $\mathcal{U}(\mathbf{x}_i) = \frac{\alpha(\mathbf{x}_i; \mathcal{A})}{\sum_{j=1}^M \alpha(\mathbf{x}_j; \mathcal{A})}$ (Zhan et al., 2022b). In our experiments, we use ENT as our basic AL sampling strategy, and the acquisition function is $\alpha_{\text{ENT}}(\mathbf{x}) = -\sum_{k=1}^K p_\theta(y = k|\mathbf{x}) \log p_\theta(y = k|\mathbf{x}))$, where $p_\theta(y|\mathbf{x})$ is the posterior class probability using classifier $f_\theta(\mathbf{x})$. Our framework is flexible because it can incorporate various AL sampling strategies if their acquisition function can convert to a querying density function. In general, AL methods that explicitly provide per-sample scores (e.g., class prediction uncertainty) or inherently provide pair-wise rankings among the unlabeled pool (e.g., $k$-Means) can convert to a querying density. Thus, many uncertainty-based measures like QBC (Seung et al., 1992), BALD (Houlsby et al., 2011; Gal et al., 2017), and Loss Prediction Loss (LPL) (Yoo & Kweon, 2019) are applicable since they explicitly provide uncertainty information per sample. Also, $k$-Means and Core-Set (Sener & Savarese, 2018) approaches are applicable since they provide pair-wise similarity information for ranking. On the other hand, AL

---

[1]In our implementation, the label of OOD data is $-1$.

approaches that only provide overall scores for the candidate subsets, such as Determinantal Point Processes (Bıyık et al., 2019; Zhan et al., 2021a), are unable to be adopted in our framework.

## 3.3 In-Distribution Confidence Score via Mahalanobis Distance

Intuitively, ID unlabeled data can be distinguished from OOD unlabeled data since the ID unlabeled data should be closer to the ID labeled data ($\mathcal{D}_l$) in the feature space. One possible solution is to calculate the minimum distance between an unlabeled sample and the labeled data of its predicted pseudo label provided by $f(\theta)$ (Du et al., 2021). If this minimum distance is large, the unlabeled sample will likely be OOD and vice versa. However, calculating pair-wise distances between all labeled and unlabeled data is computationally expensive. A more efficient method is to summarize the labeled ID data via a data distribution (probability density function) and then compute the distances of the unlabeled data to the ID distribution. Since we adopt GMMs to represent the data distribution, our ID confidence score is based on Mahalanobis distance, as motivated by (Lee et al., 2018). AL for classical ML tasks focuses on training basic learner $f_\theta(\mathbf{x})$ with fixed feature representations, while AL for DL jointly optimizes the feature representation $\mathcal{X}$ and classifier $f_\theta(\mathbf{x})$ simultaneously (Ren et al., 2021). Considering that these differences can influence the data distributions and further influence the Mahalanobis distance calculations, we adopt different settings for classical ML and DL.

**Classical ML tasks.**   We represent the data distribution estimated from $\mathcal{D}_l$ with GMMs. Specifically, since we have labels of $\mathcal{D}_l$, we represent each class $k$ with a class-conditional GMM,

$$p_{\text{GMM}}(\mathbf{x}|y=k) = \sum_{c=1}^{C_k} \pi_c^{(k)} \mathcal{N}(\mathbf{x}|\mu_c^{(k)}, \Sigma_c^{(k)}), \tag{2}$$

where $(\pi_c^{(k)}, \mu_c^{(k)}, \Sigma_c^{(k)})$ are the mixing coefficient, mean vector and covariance matrix of the $c$-th Gaussian component with the $k$-th class, and $\mathcal{N}(\mathbf{x}|\mu, \Sigma)$ is a multivariate Gaussian distribution with mean $\mu$ and covariance $\Sigma$. The class-conditional GMM for class $k$ is estimated from the labeled data for class $k$, $\mathcal{D}_l^{(k)} = \{\mathbf{x}_i|y_i = k\}$ using a maximum likelihood estimation (Reynolds, 2009) and the EM algorithm (Dempster et al., 1977). We then define the ID confidence score $\mathcal{M}(\mathbf{x})$ as the Mahalanobis distance between the unlabeled sample and its closest Gaussian component in any class,

$$\mathcal{M}_{\text{ML}}(\mathbf{x}) = \max_k \max_c -||\mathbf{x} - \mu_c^{(k)}||^2_{\Sigma_c^{(k)}}. \tag{3}$$

**DL tasks.**   For DL, we follow (Lee et al., 2018) to calculate $\mathcal{M}(\mathbf{x})$, since it makes good use of both low- and high-level features generated by the deep neural network (DNN) and also applies calibration techniques, e.g., input-preprocessing (Liang et al., 2018) and feature ensembles Lee et al. (2018) to improve the OOD detection capability. Specifically, denote $f_\ell(\mathbf{x})$ as the output of the $\ell$-th layer of a DNN for input $\mathbf{x}$. A class-conditional Gaussian distribution with shared covariance is estimated for each layer and for each class $k$ by calculating the empirical mean and *shared* covariance $(\hat{\mu}_\ell^{(k)}, \hat{\Sigma}_\ell)$:

$$\hat{\mu}_\ell^{(k)} = \frac{1}{|\mathcal{D}_l^{(k)}|} \sum_{\mathbf{x} \in \mathcal{D}_l^{(k)}} f_\ell(\mathbf{x}), \quad \hat{\Sigma}_\ell = \frac{1}{|\mathcal{D}_l|} \sum_k \sum_{\mathbf{x} \in \mathcal{D}_l^{(k)}} (f_\ell(\mathbf{x}) - \hat{\mu}_\ell^{(k)})(f_\ell(\mathbf{x}) - \hat{\mu}_\ell^{(k)})^T. \tag{4}$$

Next we find the closest class for each layer $\ell$ via $\hat{k}_\ell = \arg\min_k ||f_\ell(\mathbf{x}) - \hat{\mu}_\ell^{(k)}||^2_{\hat{\Sigma}_\ell}$. To make the ID and OOD data more separable, an input pre-processing step is applied by adding a small perturbation to $\mathbf{x}$, $\hat{\mathbf{x}} = \mathbf{x} + \varepsilon \text{sign}(\nabla_\mathbf{x} ||f_\ell(\mathbf{x}) - \hat{\mu}_\ell^{(\hat{k}_\ell)}||^2_{\hat{\Sigma}_\ell})$, where $\varepsilon$ controls the perturbation magnitude[2]. Finally, the ID confidence score is the average Mahalanobis distance for $\hat{\mathbf{x}}$ over the $L$ layers,

$$\mathcal{M}_{\text{DL}}(\mathbf{x}) = \frac{1}{L} \sum_{\ell=1}^{L} \max_k -||f_\ell(\hat{\mathbf{x}}) - \hat{\mu}_\ell^{(k)}||^2_{\hat{\Sigma}_\ell}. \tag{5}$$

## 3.4 Monte-Carlo Pareto Optimization Active Learning

Our framework for AL for OOD data contains two criteria, AL and ID confidence score. As discussed in Section 2, weighted-sum and multi(two)-stage optimization are widely used in multiple-criteria AL problems. However, both have drawbacks when applied to AL for the OOD data scenario. In weighted-sum optimization, the objective functions are summed up with weight $\eta$:

---

[2]In our experiments we follow (Lee et al., 2018) and set $\varepsilon = 0.01$.

$\alpha_{\text{WeightedSum}}(\mathbf{x}) = \eta \mathcal{U}(\mathbf{x}) + (1 - \eta)\mathcal{M}(\mathbf{x})$, e.g., as adopted by CCAL (Du et al., 2021). This introduces an extra trade-off parameter $\eta$ for tuning, but the true proportion of OOD data in $\mathcal{D}_u$ is unknown; thus, we cannot tell which criterion is more important. Furthermore, due to the lack of data, there is not enough validation data for properly tuning $\eta$. For two-stage optimization (Haneveld & van der Vlerk, 1999; Ahmed et al., 2004; Bindewald et al., 2020; Salo et al., 2022), a subset of possible ID samples is first selected with threshold $\mathcal{M}(\mathbf{x}) < \delta$, and then the most informative $b$ samples in the subset are selected by maximizing $\mathcal{U}(\mathbf{x})$. However, suppose $\delta$ has not been properly selected. In that case, we might i) select OOD samples in the first stage (when $\delta$ is too large), and these OOD samples will be more likely to be selected first due to their higher AL score in the second stage; ii) select ID samples that are close to existing samples (when $\delta$ is too small), and due to the natural conflict between the two criteria, the resulting subset will be non-informative, thus influencing the AL performance (Zhao et al., 2019).

Considering this contradiction between AL and OOD criteria, developing a combined optimization strategy that does not require manual tuning of this trade-off is vital. Therefore, we propose POAL for balancing $\mathcal{U}(\mathbf{x})$ and $\mathcal{M}(\mathbf{x})$ automatically without requiring trade-off parameters. Consider one iteration of the AL process, where we must select $b$ samples from $\mathcal{D}_u$. The search space size is the number of combinations $M$ choose $b$, $\mathcal{C}(M, b)$, and the search is an NP-hard problem in general. Inspired by (Qian et al., 2015), we use a binary vector representation of candidate subset: $\mathbf{s} \in \{0, 1\}^M$, where $s_i = 1$ represents $i$-th sample is selected, and $s_i = 0$ otherwise. For the unlabeled set $\mathcal{D}_u$, we denote the vector of AL scores as $\mathbf{u} = [\mathcal{U}(\mathbf{x}_i)]_{i=1}^M$ and ID confidence scores as $\mathbf{m} = [\mathcal{M}(\mathbf{x}_i)]_{i=1}^M$. The two criteria scores for the subset $\mathbf{s}$ are then computed as $o_U(\mathbf{s}) = \mathbf{s}^T \mathbf{u}$ and $o_M(\mathbf{s}) = \mathbf{s}^T \mathbf{m}$.[3] The ranking relationships between two candidate subsets $\mathbf{s}$ and $\mathbf{s}'$ are as follows:

- $\mathbf{s}' \preceq \mathbf{s}$ denotes that $\mathbf{s}'$ is *dominated* by $\mathbf{s}$, such that both scores for $\mathbf{s}'$ are no better than those of $\mathbf{s}$, i.e., $o_U(\mathbf{s}') \leq o_U(\mathbf{s})$ and $o_M(\mathbf{s}') \leq o_M(\mathbf{s})$.
- $\mathbf{s}' \prec \mathbf{s}$ denotes that $\mathbf{s}'$ is *strictly dominated* by $\mathbf{s}$, such that $\mathbf{s}'$ has one strictly smaller score (e.g., $o_U(\mathbf{s}') < o_U(\mathbf{s})$) and one score that is not better (e.g., $o_M(\mathbf{s}') \leq o_M(\mathbf{s})$).
- $\mathbf{s}$ and $\mathbf{s}'$ are *incomparable* if both $\mathbf{s}$ is not *dominated* by $\mathbf{s}'$ and $\mathbf{s}'$ is not *dominated* by $\mathbf{s}$.

Our goal (see Eq. 1) is to find optimal subset solution $\mathbf{s}$ that dominates the remaining subset solutions with $\sum s_i' = b$. The large search space makes traversing all possible subset solutions impossible. Thus we propose Monte-Carlo POAL for fixed-size subset selection. Monte-Carlo POAL iteratively generates a candidate solution $\mathbf{s}$ at random, and checks it against the current Pareto set $\mathcal{P} = \{\mathbf{s}_1, \mathbf{s}_2, \cdots\}$. If there are no candidate solution in $\mathcal{P}$ that *strictly dominates* $\mathbf{s}$, then $\mathbf{s}$ is added to $\mathcal{P}$ and all candidate solutions in $\mathcal{P}$ that are *dominated* by $\mathbf{s}$ are removed. In this way, a Pareto set $\mathcal{P}$ is maintained to include the solution(s) that dominate all other candidate solutions so far, while the solution(s) in $\mathcal{P}$ are incomparable with each other[4].

**Early-stopping.** We adopt an early-stopping strategy to automatically terminate POAL when there is no significant change in $\mathcal{P}$ after many successive iterations, which indicates that *a randomly generated* $\mathbf{s}$ *has little probability of changing* $\mathcal{P}$ since most non-dominated solutions are included in $\mathcal{P}$ (Saxena et al., 2016). Firstly, we need to define the difference between two Pareto sets, $\mathcal{P}$ and $\mathcal{P}'$. We represent each candidate solution $\mathbf{s}$ as a 2-dimensional feature vector, $v(\mathbf{s}) = (o_U(\mathbf{s}), o_M(\mathbf{s}))$. The two Pareto sets can then be compared via their feature vector distributions, $\mathbf{V}(\mathcal{P}) = \{v(\mathbf{s})\}_{\mathbf{s} \in \mathcal{P}}$ and $\mathbf{V}(\mathcal{P}') = \{v(\mathbf{s}')\}_{\mathbf{s}' \in \mathcal{P}'}$. We employed the maximum mean discrepancy (MMD) (Gretton et al., 2006) to measure the difference:

$$\text{MMD}(\mathcal{P}, \mathcal{P}') = \left\| \frac{1}{|\mathcal{P}|} \sum_{\mathbf{s} \in \mathcal{P}} v(\mathbf{s}) - \frac{1}{|\mathcal{P}'|} \sum_{\mathbf{s}' \in \mathcal{P}'} v(\mathbf{s}') \right\|_{\mathcal{H}}. \tag{6}$$

MMD is based on embedding probabilities in reproducing kernel Hilbert space $\mathcal{H}$, and here we use the RBF kernel with various bandwidths (i.e., multiple kernels). At iteration $t$, we compute the mean and standard deviation (SD) of the MMD scores in a sliding window of the previous $s_w$ iterations,

$$M_t = \frac{1}{s_w} \sum_{i=t-s_w+1}^{t} \text{MMD}(\mathcal{P}_{i-1}, \mathcal{P}_i), \quad S_t = \frac{1}{s_w} \sum_{i=t-s_w+1}^{t} (\text{MMD}(\mathcal{P}_{i-1}, \mathcal{P}_i) - M_t)^2. \tag{7}$$

If there is no significant difference in the mean and SD after several iterations, e.g., $M_t$ and $S_t$ do not change within two decimal points, then we assume that $\mathcal{P}$ has converged, and we stop iterating.

---

[3]To keep the two criteria consistent, we normalize the ID confidence scores: $\mathbf{m} \leftarrow \max(\mathbf{m}) - \mathbf{m}$.
[4]POAL is inspired by POSS (Qian et al., 2015), but have essential differences, see Appendix B for details.

**Final selection.**     After obtaining the converged set $\mathcal{P}$, we need to pick one $\mathbf{s} \in \mathcal{P}$ as the final solution (i.e., final subset for querying). Selecting a final solution is still an open question in multi-objective optimization (see Appendix C.3 for more details).

$$\mathbf{s}^* = \arg\min_{\mathbf{s} \in \mathcal{P}} \sum_{\mathbf{s}' \in \mathcal{P}} \|\mathbf{s}' - \mathbf{s}\|^T = \arg\max_{\mathbf{s} \in \mathcal{P}} \sum_{\mathbf{s}' \in \mathcal{P}} \mathbf{s}^T \mathbf{s}', \tag{8}$$

where we use the property $\mathbf{s}^T \mathbf{s} = \mathbf{s}'^T \mathbf{s}' = b$. In (8), we select the solution with a maximum intersection with other non-dominated solutions, i.e., we find the solution whose selected samples are commonly used in all other solutions in $\mathcal{P}$. Each solution in $\mathcal{P}$ represents a particular ID-OOD trade-off since an appropriate choice of weight $\eta$ in the weighted-sum objective will lead to its selection. Thus, the maximum intersection operation selects the samples that are *common to all settings* of the ID-OOD trade-off. In this sense, POAL bypasses the tuning of ID-OOD trade-off hyper-parameters by selecting samples that work well for all trade-offs.

**Pre-selection for large-scale data.**     We next consider using POAL on large datasets. The search space of size $\mathcal{C}(M, b)$ is huge for a large unlabeled data pool. We propose an *optional* pre-selection technique to decrease the search space size. We select an optimistic subset $\mathcal{D}_{\text{sub}}$ from $\mathcal{D}_u$, based on the dominance relationships between different pairs of samples from the original unlabeled data pool. Firstly, an initial Pareto front $\mathcal{P}_{\text{pre}}$, a set of non-dominated data points, is selected according to two objectives $\mathcal{U}(\mathbf{x})$ and $\mathcal{M}(\mathbf{x})$. Since the size of $\mathcal{P}_{\text{pre}}$ is not fixed and might not meet our requirement of the minimum pre-selected subset size $s_m$, we iteratively update $\mathcal{D}_{\text{sub}}$ by firstly adding the data points in $\mathcal{P}_{\text{pre}}$ to $\mathcal{D}_{\text{sub}}$, and then excluding $\mathcal{D}_{\text{sub}}$ from $\mathcal{D}_u$ for the next round of selection. The iteration terminates when $|\mathcal{D}_{\text{sub}}| \geq s_m$. The pre-selected subset size $s_m$ is set according to personal requirements (i.e., the computing resources and time budget). In our experiment, we set $s_m = 6b$.

We summarized POAL with early-stopping and pre-selection in Algorithms 1 and 2 in the Appendix A. The computational analysis is in Appendix D. More discussions of POAL are in Appendix G.

## 4 EXPERIMENTS

In the experiments we aim to: 1) evaluate the effectiveness of POAL on both classical ML and DL tasks under various scales of OOD data scenarios; 2) compare different multi-objective optimization strategies, i.e., weighted-sum, two-stage, and Pareto optimization; 3) observe how OOD data influence typical AL methods as discussed in Section 2.

### 4.1 EXPERIMENTAL DESIGN

**Datasets.**     For classical ML tasks, we use pre-processed data from LIBSVM (Chang & Lin, 2011):

- Synthetic data: *EX8* uses *EX8a* as ID data and *EX8b* as OOD data (Ng, 2008).
- Real-life data: *Vowel* (Asuncion & Newman, 2007; Aggarwal & Sathe, 2015) has 11 classes, and we use 7 classes as ID data and the remaining for OOD data. *Letter* (Frey & Slate, 1991; Asuncion & Newman, 2007) has 26 classes, and we use 10 (*a-j*) as ID data and the remaining 16 classes (*k-z*) as OOD data. We also construct 16 datasets with increasing ID:OOD ratios, denoted as *letter(a-k)*, *letter(a-l)*,..., *letter(a-z)*.

For DL tasks, we adopt the following image datasets:

- *CIFAR10* (Krizhevsky et al., 2009) has 10 classes, and we construct two datasets: *CIFAR10-04* splits the classes with ID:OOD ratio of 6:4, and *CIFAR10-06* splits data with ratio as 4:6.
- *CIFAR100* (Krizhevsky et al., 2009) has 100 classes, and we construct *CIFAR100-04* and *CIFAR100-06* using the same ratios as *CIFAR10*.

Appendix E.2 shows visualizations of the ID and OOD data in these datasets.

**Baselines.**     Our work helps existing AL methods select more informative ID data samples while preventing OOD data selection. In our experiments, we adopt ENT as our basic AL sampling strategy for POAL framework. We compare against two baseline methods considering OOD data: CCAL (Du et al., 2021) and SIMILAR (with FLVMI submodular function) (Kothawade et al., 2021). We compare against five normal AL methods without OOD detection: ENT, BALD (Gal et al., 2017)

and LPL (Yoo & Kweon, 2019) as uncertainty-based measures; $k$-Means as representative-based measure and BADGE (Ash et al., 2020) as combined strategy. To show the effectiveness of our MC Pareto optimization, we compared it against two alternative combination strategies: weighted sum optimization (WeightedSum) using weights $\eta = \{0.2, 0.5, 0.8\}$ and two-stage optimization (TwoStage) using threshold $\delta = \texttt{mean}(\mathbf{m})$. Additionally, we add random sampling (RAND) and Mahalanobis distance (MAHA) as baselines. Finally, we report the oracle results of ENT where only ID data is selected first (IDEAL-ENT), which serves as ideal performance of POAL.

**Implementation details.** The training/test split of the datasets is fixed in the experiments, while the initial label set and the unlabeled set is randomly generated from the training set. Experiments are repeated 100 times for classical ML tasks and 3 for DL tasks. We use the same basic classifier for each dataset (details in Appendix E, Table 1). To evaluate model performance, we measure accuracy and plot accuracy vs. budget curves to present the performance change with increasing labeled samples. To calculate average performance, we compute the area under the accuracy-budget curve (AUBC) (Zhan et al., 2021b), with higher values reflecting better overall performance under varying budgets. More details about the experiments are in Appendix E, including data information, dataset splits, ID : OOD ratios, basic learner settings, and AL hyper-parameters (budget $B$, batch size $b$, and initial label set size), etc.

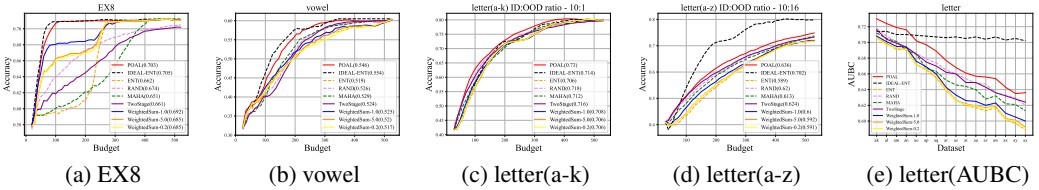

| (a) EX8 | (b) vowel | (c) letter(a-k) | (d) letter(a-z) | (e) letter(AUBC) |

Figure 3: (a)-(d) are accuracy vs. budget curves for classical ML tasks. The AUBC performances are shown in parentheses in the legend. To observe the effect of increasing ID:OOD ratio on *letter* datasets, we plot AUBC vs. dataset curves in (e). The complete figures (more *letter* datasets) are shown in Appendix F.1, Figs. 6 and 7.

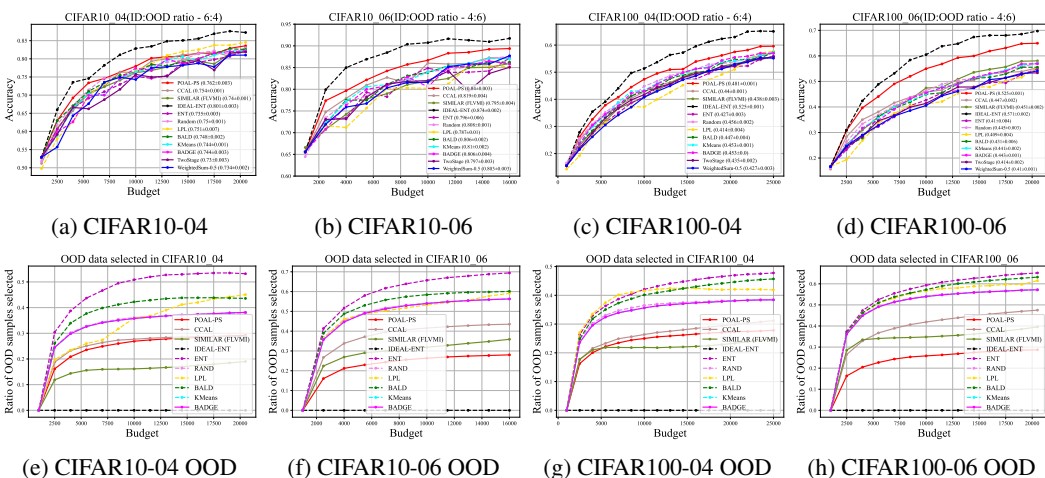

| (a) CIFAR10-04 | (b) CIFAR10-06 | (c) CIFAR100-04 | (d) CIFAR100-06 |

| (e) CIFAR10-04 OOD | (f) CIFAR10-06 OOD | (g) CIFAR100-04 OOD | (h) CIFAR100-06 OOD |

Figure 4: Results of POAL with **P**re-**S**election on DL datasets. (a-d) are accuracy vs. budget curves. We present the mean and standard deviation (in parentehsis) of AUBC. (e-h) plot the ratio of OOD samples selected.

## 4.2 RESULT ANALYSIS

**Overall performance.** We present the overall performance of classical ML and DL tasks in Fig. 3 and Fig. 4, respectively. Considering each dataset (see data visualization in App. E.2), the ideal performance of POAL is shown on the synthetic dataset *EX8* (Fig. 3a), the ID data is non-linearly separable, and the OOD data is far from ID data. These properties make the calculation of $\mathcal{M}$ very accurate. Meanwhile, OOD and ID data close to the decision boundaries have high entropy scores. The curve of our POAL is the closest to IDEAL-ENT, which indicates that POAL selects the most informative ID data while excluding the OOD samples. For real-life classical ML datasets like *vowel*

and *letter* (Fig. 3b-e), POAL also demonstrates its superiority. In *letter*, we fixed the ID data (letters *a-j*) and gradually increase the OOD data (letters *k-z*). From Figure 3e, we observe that except for IDEAL-ENT, all methods are influenced by increasing OOD data, e.g., the AUBC of RAND is $0.718$ in Fig. 3c, $0.62$ in Fig. 3d. Although our method is also influenced by increasing OOD data, it is still superior to all baselines. POAL also works on large-scale data with DL tasks. Fig. 4 shows the accuracy-budget curves (along with the AUBC (acc) values) for *CIFAR10* and *CIFAR100* with ID:OOD ratios 6:4 and 4:6, together with the number of OOD samples selected during AL processes. POAL outperforms both uncertainty-, representative/diversity-based and combined AL baselines like LPL, BALD, $k$-Means and BADGE on all tasks. We next analyze various aspects of the experiments.

**POAL vs. SIMILAR and CCAL.** These three methods consider OOD data when selecting samples. As shown in Fig. 4e-h, the three methods effectively reduce the amount of OOD data that is selected. POAL outperforms CCAL and SIMILAR (FLVMI) by large margins on more challenging situations (i.e., ID:OOD ratio is 4:6), POAL selects fewer OOD samples and achieves better AUBC performance (e.g., the AUBC value of POAL on *CIFAR100-06* is $0.525$, while the highest baseline SIMILAR is $0.451$). SIMILAR selects data samples close to labeled ID data and dissimilar to OOD data. It prevents selecting the OOD data, but the selected ID data may not be informative enough. CCAL utilized weighted-sum optimization to balance the informative and distinctive selections, and the trade-off parameter heavily affects its performance (also see Section 4.3 in (Du et al., 2021)). Note that SIMILAR has multiple implementations with various submodular mutual information functions, and to provide a fair comparison, we also compared the most effective submodular functions (FLCMI) on the down-sampled *CIFAR10* dataset, as shown in the Appendix F.3.

**POAL vs. other multi-objective optimization strategies.** We compared our Pareto optimization against weighted-sum, two-stage optimization, and each single objective, i.e., ENT and MAHA. Both ENT and MAHA are ineffective when used as a single selection strategy under OOD scenarios – in Fig 3a, ENT always selects OOD data with higher entropy until all OOD data are selected (*EX8* contains 210 OOD samples), and it performs even worse than RAND. MAHA prefers to select ID samples first (see Appendix Fig. 7), but the data samples with smaller Mahalanobis distance are also easily classified and far from the decision boundary, and thus not informative. POAL outperforms the other combination strategies, WeightedSum for different $\eta$ values and TwoStage, on all tasks. This demonstrates that joint Pareto optimization better handles the two conflicting criteria.

**POAL vs. normal AL.** We run experiments on normal AL methods, including uncertainty-based (ENT, LPL and BALD), diversity-based ($k$-Means) and combined methods (BADGE). Although LPL performs fairly well on standard *CIFAR10* and *CIFAR100* datasets (Yoo & Kweon, 2019; Zhan et al., 2022a), in OOD scenarios, OOD data have larger predicted loss values and thus influence the selection. Similar phenomena were observed with BALD and ENT. The performances of $k$-Means are close to RAND since both methods sampled ID/OOD data with the same ratio. Although $k$-Means will not select OOD samples first like uncertainty-based measures, it still failed to provide comparable performances. POAL performs well by selecting fewer OOD samples during the AL iterations, as shown in Fig. 4(e-h). BADGE performed similar to $k$-Means. The first stage of BADGE calculates the gradient embedding per sample, and use $k$-Means++ for clustering. These operations results in both high informative ID and OOD data are mixed together with high gradient values, therefore, BADGE failed to distinguish ID/OOD data. In DL tasks, we adopted pre-selection to deal with large-scale datasets. Although it makes the search space smaller, i.e., changes the global solutions to local solutions, POAL-PS still performs significantly better than baseline AL methods.

Additional experiments including sensitivity analysis of $s_w$, POAL with other AL methods, more multi-objective optimization methods, POAL vs. SIMILAR (FLCMI) are presented in Appendix F.

## 5 CONCLUSION

In this paper, we proposed an effective and flexible Monte Carlo POAL framework for improving AL when encountering OOD data. Our POAL framework i) incorporates various AL sampling strategies; ii) handles large-scale datasets; iii) supports batch-mode AL with fixed batch size; iv) does not require tuning a trade-off hyper-parameter; and v) works well on both classical ML and DL tasks. Future work could improve the OOD detection criterion by introducing OOD-specific feature representations and other techniques for better distinguishing ID/OOD distributions.

## 6 Ethic Statement

There is a legitimate concern that POAL with pre-selection strategies (**POAL-PS**) on large-scale datasets that would induce bias problems. Since the Pre-Selection technique would firstly filter some unlabeled data samples in one AL iteration, and thus have no probability of being selected. Future work should concern more about pre-selection techniques. What kind of high informative ID data samples may we miss? Is the pre-selected subset enough representative?

## 7 Reproducibility Statement

For models, we provide an implementation of POAL in supplementary material, including the classical ML task and DL task versions. For datasets, all datasets we employed in this paper are public datasets.

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

# Appendix

## Table of Contents

## A    PSEUDO CODES OF POAL

We summarize the POAL with an early stopping technique and pre-selection operation in Algorithm 1 and Algorithm 2, respectively. In Algorithm 1. The implementation of MMD is the RBF kernel with various bandwidths with open-source implementation[5].

## B    MORE DISCUSSIONS OF POAL AND POSS

In this section, we mainly introduce Pareto Optimization for Subset Selection POSS (Qian et al., 2015) and how our POAL is inspired from POSS. We then discuss the major difference between POAL and POSS.

---

[5]The PyTorch implementation of MMD is: `https://github.com/easezyc/deep-transfer-learning/blob/master/MUDA/MFSAN/MFSAN_3src/mmd.py`

---

**Algorithm 1** Monte-Carlo POAL with early stopping under OOD data scenarios.

---

**Require:** Unlabeled data pool $\mathcal{D}_u$ with size $M$, criteria $\mathcal{U}(\mathbf{x})$ and $\mathcal{M}(\mathbf{x})$, batch size $b$, maximum repeat time $T$, population interval $p_{\text{inv}}$, sliding window size $s_w$ and final decision function $\mathcal{F}$.
**Ensure:** A subset $\mathcal{D}_s$ of $\mathcal{D}_u$ where $|\mathcal{D}_s| = b$.
1: Let $\mathbf{s} = \{0\}^M$, $\mathcal{P} = \{\mathbf{s}\}$ and $t = 1$.
2: **for** $t$ in $1, ..., T$ **do**
3:    Generate a random solution $\mathbf{s}$ with condition $\sum s_i = b$.
4:    **if** $\nexists \mathbf{z} \in \mathcal{P}$ such that $\mathbf{s} \prec \mathbf{z}$ **then**
5:       $Q = \{\mathbf{z} \in \mathcal{P} | \mathbf{z} \preceq \mathbf{s}\}$.
6:       $\mathcal{P} = (\mathcal{P} \setminus Q) \cup \{\mathbf{s}\}$.
7:    **end if**
      {Terminate the loop early if the pareto set $\mathcal{P}$ converged.}
8:    **if** $t \neq 0$ **then**
9:       Calculate $\text{MMD}_t$ according to Equation 5.
10:      **if** $\text{MOD}(t, p_{\text{inv}}) = 0$ **then**
11:         Calculate $M_t$ and $S_t$ with interval $p_{\text{inv}}$ according to Equation 6 respectively.
12:         $\hat{M}_t = \text{ROUND}(M_t, 2)$, $\hat{S}_t = \text{ROUND}(S_t, 2)$.
13:         **if** $\hat{M}_t = \hat{M}_{t-1} = ... = \hat{M}_{t-s_w}$ and $\hat{S}_t = \hat{S}_{t-1} = ... = \hat{S}_{t-s_w}$ **then**
14:            **Break**
15:         **end if**
16:      **end if**
17:    **end if**
18: **end for**
19: **return** $\arg\max_{\mathbf{s} \in \mathcal{P}} \mathcal{F}(\mathbf{s})$.

---

**Algorithm 2** Pre-selecting technique on large-scale datasets.

---

**Require:** Unlabeled data pool $\mathcal{D}_u$, criteria $\mathcal{U}$ and $\mathcal{M}$, minimum size of subset $S_m$ of pre-selection.
**Ensure:** A pre-selected subset $\mathcal{D}_{\text{sub}}$ of $\mathcal{D}_u$ with $|\mathcal{D}_{\text{sub}}| \geq S_m$.
1: Let $\mathcal{D}_{\text{sub}} = \emptyset$. $i = 0$.
2: **while** $|\mathcal{D}_{\text{sub}}| < s_m$ **do**
3:    Let $\mathcal{D}_p = \emptyset$.
4:    **for** $i$ in $0, ..., |\mathcal{D}_u|$ **do**
5:       **if** $\nexists \mathbf{x} \in \mathcal{D}_p$ such that $\mathcal{D}_u^i \prec \mathbf{x}$ **then**
6:          $Q = \{\mathbf{x} \in \mathcal{D}_p | \mathbf{x} \preceq \mathcal{D}_u^i\}$.
7:          $\mathcal{D}_p = (\mathcal{D}_p \setminus Q) \cup \{\mathcal{D}_u^i\}$.
8:       **end if**
9:    **end for**
10:    $\mathcal{D}_u = \mathcal{D}_u \setminus \mathcal{D}_p$.
11:    $\mathcal{D}_{\text{sub}} = \mathcal{D}_{\text{sub}} \cup \mathcal{D}_p$.
12: **end while**
13: **return** $\mathcal{D}_{\text{sub}}$.

---

We briefly introduce POSS first. POSS is originally applied to the subset selection problem, which is defined as follows:

$$\arg\min_{S \subseteq V} f(S) \quad s.t. \quad |S| \leq k,$$

where $V = \{X_1, ..., X_n\}$ is a set of variables, $f$ is a criterion function and $k$ is a positive integer. The subset selection problem aims to select a subset $S \subseteq V$ such that $f$ is optimized with the constraint $|S| \leq k$, where $|\cdot|$ denotes the size of a set. The subset selection problem is NP-hard in general (Davis et al., 1997).

POSS solves the subset selection by separating the problem into two objectives, namely optimizing the criterion function and minimizing the subset size. Usually, the two objectives are conflicting. Thus the problem is transformed into a multi-objective optimization problem:

$$\min_{S \subseteq V} \quad (f(S), |S|).$$

Due to different trade-offs of objectives, multi-objective optimization algorithms need to find a Pareto set $\mathcal{P}$ containing Pareto optimal solutions. Specifically, POSS first initializes $\mathcal{P}$ with a random solution and then selects a solution from $\mathcal{P}$ to generate a new solution by flipping each bit with probability $1/n$ ($n$ is the number of variables). The new solution will be added to $\mathcal{P}$ if it is not strictly dominated by any solution in $\mathcal{P}$, and the solutions in $\mathcal{P}$ that are dominated by the new solution will also be excluded. POSS repeats this iteration for $T$ times, and it is proved to achieve the optimal approximation guarantee of $(1 - 1/e)$ with $E[T] \leq 2ek^2n$ expected running time.

Our Monte-Carlo POAL is inspired by POSS, but there are two essential differences, as follows:

Firstly, although POSS is in the form of bi-objective optimization, it supports only one criterion function, e.g., just $\mathcal{U}(\mathbf{x})$, while the other criterion is that the subset size does not exceed $b$ (i.e., $\sum_i s_i \leq b$). However, our POAL needs to solve two different criterion functions, which is a much harder problem.

Secondly, POSS solves the problem with the constraint $|S| \leq k$. POSS generates new solutions by flipping the bits of the solutions from the previous iterations, i.e., solutions in $\mathcal{P}$. This operation may change the subset size, violating our setting of fixed-size solutions. Thus, we adopt the Monte-Carlo approach to generate fixed-size solutions unrelated to the solutions from the previous iterations. However, our POAL needs fixed-size subset solutions, which means the search space is different.

Due to these differences, we proposed Monte Carlo POAL for our AL under OOD data scenarios tasks. We randomly generate candidate solutions with fixed batch sizes. It 1) controls all candidate solutions in $\mathcal{P}$ have the same fixed size; 2) compared with POSS, Monte Carlo POAL does not depend on previous selections, which keeps the method free from initialization problems.

POSS cannot satisfy our requirements. Due to these differences in how candidate solutions are generated and the number of criteria, the theoretical bound on the number of iterations $T$ needed for convergence of POSS (Qian et al., 2015) cannot be applied to our Monte-Carlo POAL method. Thus, when should the iterations of our Monte-Carlo POAL terminate? We noticed in previous research (Qian et al., 2017) that the Pareto set empirically converges much faster than $E[T]$ (see Fig. 2 in (Qian et al., 2017)). Therefore, we propose an *early-stopping* technique, which has been introduced in Section 3.4.

## C  RELATED WORK

This section is an extension of the Section 2. We next introduce two methods that consider AL under OOD data scenarios and the relationship/difference between the scenarios of OOD AL and AL with biased data. We also discussed the final selection problem in Multi-objective optimization.

### C.1  AL UNDER OOD DATA SCENARIOS

We discuss the related work that involves AL under OOD data scenarios. As mentioned in the main paper, there is little related work on AL under OOD data scenarios. To the best of our knowledge, there are two published works that discussed AL under OOD data scenarios: Contrastive Coding for Active Learning (CCAL) (Du et al., 2021) and Submodular Information Measures Based Active Learning In Realistic Scenarios (SIMILAR) (Kothawade et al., 2021). We next briefly introduce these two papers and compare them with our work.

### C.1.1  POAL VS. CCAL

Du et al. (2021) considers the class distribution mismatch problem in AL. Their goal is to select the most informative samples with matched categories. CCAL is the first work related to AL for class distribution mismatch. It proposed a contrastive coding-based method, which extracts semantic and distinctive features by contrastive learning. Semantic feature refers to category-level features that can be exploited to filter invalid samples with mismatched categories. Distinctive features describe the individual level. It is an AL task-specific feature to select the most representative and informative class-matched unlabeled data samples. CCAL achieves good performance on well-studied datasets, e.g., CIFAR10 and CIFAR100 datasets. CCAL utilized self-supervised models like SimCLR and CSI. It is an excellent idea since it will extract semantic and distinctive features compared with normal

feature representations like the output of the penultimate layer of neural networks. However, it also brings limitations, that training a self-supervised model has both high computational and timing costs, especially on large-scale data sets. Based on the release code of CCAL[6], we take 3 days to train a distinctive feature extraction model (with 700 epochs and batch size 32, on CIFAR10 dataset) on a V100 GPU. Additionally, CCAL adopted weighted-sum optimization to combine semantic and distinctive scores, whose acquisition function is as follows:

$$\alpha_{\textbf{CCAL}} = \tanh\left(k S_{\text{semantic}(\mathbf{x})-t}\right) + S_{\text{distinctive}}(\mathbf{x}),$$

where the threshold $t$ is for selectively narrowing the semantic scores of samples, and $k$ controls the slope of the $\tanh$ function. $\alpha_{\textbf{CCAL}}$ introduces two hyper-parameters for balancing the semantic and distinctive scores. As discussed in Section 4.3 in (Du et al., 2021), the choice of the two hyper-parameters will influence the final performance. The final shortcoming comes from calculating the distinctive score to obtain representativeness information. **CCAL** needs pair-wise comparison among the whole data pool, whicih time and memory-consuming on large-scale datasets.

### C.1.2    POAL VS. SIMILAR

SIMILAR is an AL framework using the previously proposed submodular information measures (SIM) as acquisition functions. Apparently, (Kothawade et al., 2021) is an application of SIM (Iyer et al., 2021; Kaushal et al., 2021). Kothawade et al. (2021) proposed many SIM variants to deal with realistic scenarios, e.g., data imbalance, rare-class, OOD, etc. AL under the OOD data scenario is a sub-task of their work. They adopted a submodular conditional mutual information (SCMI) function that best matches AL under OOD data scenario tasks, that is, Facility Location Conditional Mutual Information (FLCMI), and the function is:

$$\mathcal{I}_f(A, Q|P) = \sum_{i \in V} \max(\min(\max_{j \in A} s_{ij}, \eta \max_{j \in Q} s_{ij}) - \nu \max_{j \in P} s_{ij}, 0),$$

where they use the currently labeled OOD points as the conditioning, set $P$, and the currently labeled in-distribution (ID) points as the query set $Q$. $A$ is unlabeled data set. It jointly models the similarity between $A$ and $Q$ and their dissimilarity with $P$. The advantages and disadvantages of **SIMILAR** are both clear. However, the proposed FLCMI is memory-consuming, and it cannot handle large-scale datasets. For instance, we cannot run experiments on the full CIFAR10 dataset with one V100 GPU with 32GB memory (memory error is reported). In (Kothawade et al., 2021), they conduct OOD-related experiments on a downsampled version of CIFAR10. They downsample the dataset to size $15.6K$ (the size of the whole CIFAR10 dataset for training is $50K$).

(Kothawade et al., 2021) provided a partition trick to solve the memory-consuming problem by randomly separating the unlabeled data into several pieces and running the AL algorithm on every piece. It is effective but will hurt the final performance, especially when the number of partitions is large. We do not use the partition trick in our experiments to provide a fair comparison. We conduct a time and memory-efficient submodular function FLVMI as the baseline in main experiments (see Fig 4). FLVMI has comparable performance in the experimental part of (**?**). We also conduct experiments on the down-sampled *CIFAR10* dataset, following the same experimental settings in Kothawade et al. (2021). The function of FLVMI is:

$$\mathcal{I}_f(A, Q) = \sum_{i \in |\mathcal{D}_u|} \min(\max_{j \in A} S_{ij}, \max_{j \in Q} S_{ij}).$$

To sum up, the existing research on AL under OOD data scenarios contains the following problems to be overcome: 1) time/memory cost on large-scale datasets; 2) additional trade-off hyper-parameters for balancing different (even conflict) criteria need hand-tuning or tuned by extra validation set. We propose our POAL framework to address these issues. To solve the first problem, we propose pre-selection and early stopping techniques to reduce the search space to save time and memory cost. To solve the second problem, we apply Pareto optimization for balancing various (even conflict) objectives. Therefore, our proposed framework can more easily be adapted to various tasks.

---

[6]https://github.com/RUC-DWBI-ML/CCAL

## C.2 AL WITH DATA DISTRIBUTION SHIFT

There are similarities between AL with OOD scenarios and AL with distribution shift scenarios. Both have a gap between true underlying data distribution and the estimated distribution (estimated from labeled and unlabeled data pool). The significant difference between Al techniques under OOD data scenarios and AL techniques with distribution shift lies in the difference between OOD and distribution shift. Distribution shift (aka dataset shift) refers to the discrepancy between the data distributions of the training and testing sets (or true underlying data distribution) (Zhan et al., 2022b). It causes a principle problem during the model fitting step in AL since some regions with large densities in the unlabeled data pool may not be well represented by the labeled data. For AL with distribution shifts, some work like (Beygelzimer et al., 2009; Sawade et al., 2010; Ganti & Gray, 2012; Farquhar et al., 2021; Zhan et al., 2022b) tried first to model the discrepancy between labeled data and true underlying data distribution based on techniques like importance sampling and then train unbiased basic learner(s). Some work model the difference/discrepancy between labeled set and unlabeled set (or the full data pool) to help construct AL sampling strategies. Shui et al. (2020) utilizes the unlabeled data information by training a discriminator to distinguish labeled and unlabeled data sets. Sinha et al. (2019) learns the distribution of labeled data in a latent space using a VAE. A binary adversarial classifier (discriminator) is then trained to predict unlabeled examples. Mahmood et al. (2021) minimizes the Wasserstein distance between the unlabeled set and the set to be labeled as AL sampling strategy. de Mathelin et al. (2021) discuss AL for general loss functions under domain shift and further provide a generalization bound of the target risk involving pairwise distances between sample points based on localized discrepancy distance. Distribution shifts can be further applied to train better basic classifiers (Imberg et al., 2020; Farquhar et al., 2021; Zhan et al., 2022b). However, in AL with OOD data scenarios, OOD samples are hard to be utilized since they are useless to model training since OOD samples are not in the classes of interest of the classification task. Thus, the current research aims to detect OOD samples and avoid sampling them.

## C.3 FINAL SELECTION IN MULTI-OBJECTIVE OPTIMIZATION

Selecting a solution from a large Pareto set $\mathcal{P}$ is potentially intractable for a decision maker. It is an open question and is still being discussed in the multi-objective optimization field. Previous work (Chaudhari et al., 2010) has summarized several ways to obtain a final solution: (i) reformulate the problem as a single objective problem using additional information as required by the decision-makers, such as the relative importance or weights of the objectives, goal levels for the objectives, values functions, etc. (ii) Decision makers interact with the optimization procedure typically by specifying preferences between pairs of presented solutions. (iii) Output a representative set of non-dominated solutions approximating the Pareto front, e.g., regarding the candidate solutions as data points, perform clustering and outputting the centroid as a final solution.

Considering our tasks: AL under OOD data scenarios, (i) needs external information, e.g., which criterion is more important? Alternatively, it needs the extra trade-off parameters of the AL/OOD criteria. Nevertheless, in our work, we aim to propose a general framework. We must consider the worst situation; we have no extra information. If extra information is accessible, we can use (i) to get the final solution. (ii) needs to manually pick a solution or provide the preferences, which is also inaccessible. (iii) is the only way to select a final solution without additional information, which is why we adopt it.

## D COMPUTATIONAL COMPLEXITY ANALYSIS OF POAL

### D.1 MONTE-CARLO POAL

Our proposed Monte-Carlo POAL for fixed-size subset selection used unordered sampling with replacement. Denote the ground set $\mathcal{A}$ with size $N$. At each iteration, $B$ (batch size, $B < N$) samples are randomly selected. So the search space contains $S = C_N^B$ combinations. The worst case of POAL is to find all Pareto optimal solutions $\mathcal{P} = \{s_1, s_2, \cdots\}$ out of $S$ combinations (each solution is a subset of $B$ distinct samples out of $N$ samples). Let $|\mathcal{P}| = M$ ($M \leq S$). Hence, the computational complexity is the expected number of iterations that all of the $M$ Pareto optimal solutions have been selected at least once, which is calculated below:

At each iteration, there is a probability $\frac{S-1}{S}$ that one particular solution $s_i$ is not selected. So after $k$ iterations, there is probability $(\frac{S-1}{S})^k$ that solution $s_i$ has not been selected. Thus, the probability that solution $s_i$ has been selected at least once after $k$ iterations is $1 - (\frac{S-1}{S})^k$. For all $M$ solutions, the probability that they have all been selected at least once after $k$ iterations is $(1 - (\frac{S-1}{S})^k)^M$. To obtain the probability that these $M$ solutions have been selected at least once **exactly** after $k$ iterations (i.e., not before $k$ iterations), we need to subtract the probability that it is completed after $k-1$ iterations: $(1 - (\frac{S-1}{S})^k)^M - (1 - (\frac{S-1}{S})^{k-1})^M$. Hence, the expected number of iterations is:

$$E[T] = \sum_{k=0}^{\infty} k \cdot (1 - (\frac{S-1}{S})^k)^M - (1 - (\frac{S-1}{S})^{k-1})^M$$
$$= \sum_{k=0}^{\infty} 1 - (1 - (\frac{S-1}{S})^k)^M.$$

For small datasets, the size is tolerable, so the ground set $\mathcal{A}$ is just the unlabeled dataset $\mathcal{D}_u$ with size $n$, and the search space size $S = C_n^B$. In practice, we applied an early-stopping strategy, which can obtain a good enough solution with much fewer iterations.

### D.2 PRE-SELECTION FOR LARGE-SCALE DATASETS

We designed a pre-selection technique for large-scale datasets to reduce the search space of POAL. As introduced in the last paragraph in Section 3.4, we iteratively select Pareto optimal samples from the unlabeled data. The worst case is that all unlabeled samples are Pareto optimal samples, so the number of comparisons is $0 + 1 + 2 + \cdots + (n-1) = \frac{n^2}{2} - n$. The computational complexity of the pre-selection is $O(n^2)$.

We pre-selected $s_m = \alpha B$ samples out of $n$ unlabeled samples ($\alpha$ is set according to the computing resources and time budget, we set $\alpha = 6$ in our experiment) for the subsequent Monte-Carlo POAL. Therefore, $N = \alpha B \ll n$, and the search space size $S$ is reduced from $C_n^B$ to $C_{\alpha B}^B$. Also, with an early-stopping strategy, our POAL can achieve a good enough solution with an acceptable number of iterations in practice.

## E EXPERIMENTAL SETTINGS

### E.1 DATASETS

We summarize the datasets we adopted in our experiments in Table 1, including how to split the datasets (initial labeled data, unlabeled data pool, and test set), dataset information (number of categories, number of feature dimensions), and task-concerned information (maximum budget, batch size, numbers of repeated trials and basic learners adopted in each task). Primarily, we recorded the ID : OOD ratio in each task. We down-sampled the *letter* dataset and controlled that each category only has 50 data samples in the training set.

Here we list the licence of the datasets used in our experiments:

- *EX8a* and *EX8b* (Ng, 2008): Not listed.
- *vowel* (Asuncion & Newman, 2007; Aggarwal & Sathe, 2015; Dua & Graff, 2017): Aucune licence fournie.
- *letter* (Frey & Slate, 1991; Asuncion & Newman, 2007; Dua & Graff, 2017): Not listed.
- *CIFAR10* and *CIFAR100* (Krizhevsky et al., 2009): MIT Licence.

Table 1: Datasets used in the experiments.

| Dataset | # of ID classes | # of feature dimension | # of initial labeled set | # of un-labeled pool | # of test set | # of Maxi-mum Budget | ID : OOD Ratio | batch size $b$ | # of repeat trials | basic learner |
|---|---|---|---|---|---|---|---|---|---|---|
| EX8 | 2 | 2 | 20 | 650 | 306 | 500 | 46 : 21 | 10 | 100 | GPC |
| vowel | 7 | 10 | 25 | 503 | 294 | 500 | 336 : 192 | 10 | 100 | GPC |
| letter(a-k) | 10 | 16 | 30 | 520 | 500 | 550 | 10 : 1 | 10 | 100 | LR |
| letter(a-l) | 10 | 16 | 30 | 570 | 500 | 550 | 10 : 2 | 10 | 100 | LR |
| letter(a-m) | 10 | 16 | 30 | 620 | 500 | 550 | 10 : 3 | 10 | 100 | LR |
| letter(a-n) | 10 | 16 | 30 | 670 | 500 | 550 | 10 : 4 | 10 | 100 | LR |
| letter(a-o) | 10 | 16 | 30 | 720 | 500 | 550 | 10 : 5 | 10 | 100 | LR |
| letter(a-p) | 10 | 16 | 30 | 770 | 500 | 550 | 10 : 6 | 10 | 100 | LR |
| letter(a-q) | 10 | 16 | 30 | 720 | 500 | 550 | 10 : 7 | 10 | 100 | LR |
| letter(a-r) | 10 | 16 | 30 | 870 | 500 | 550 | 10 : 8 | 10 | 100 | LR |
| letter(a-s) | 10 | 16 | 30 | 920 | 500 | 550 | 10 : 9 | 10 | 100 | LR |
| letter(a-t) | 10 | 16 | 30 | 970 | 500 | 550 | 10 : 10 | 10 | 100 | LR |
| letter(a-u) | 10 | 16 | 30 | 1020 | 500 | 550 | 10 : 11 | 10 | 100 | LR |
| letter(a-v) | 10 | 16 | 30 | 1070 | 500 | 550 | 10 : 12 | 10 | 100 | LR |
| letter(a-w) | 10 | 16 | 30 | 1120 | 500 | 550 | 10 : 13 | 10 | 100 | LR |
| letter(a-x) | 10 | 16 | 30 | 1170 | 500 | 550 | 10 : 14 | 10 | 100 | LR |
| letter(a-y) | 10 | 16 | 30 | 1220 | 500 | 550 | 10 : 15 | 10 | 100 | LR |
| letter(a-z) | 10 | 16 | 30 | 1270 | 500 | 550 | 10 : 16 | 10 | 100 | LR |
| CIFAR10-04 | 6 | $32 \times 32 \times 3$ | 1000 | 49000 | 6000 | 20000 | 6 : 4 | 500 | 3 | ResNet18 |
| CIFAR10-06 | 4 | $32 \times 32 \times 3$ | 1000 | 49000 | 4000 | 15000 | 4 : 6 | 500 | 3 | ResNet18 |
| CIFAR100-04 | 60 | $32 \times 32 \times 3$ | 1000 | 49000 | 6000 | 25000 | 6 : 4 | 500 | 3 | ResNet18 |
| CIFAR100-06 | 40 | $32 \times 32 \times 3$ | 1000 | 49000 | 4000 | 20000 | 4 : 10 | 500 | 3 | ResNet18 |
| Down-sampled CIFAR10 | 8 | $32 \times 32 \times 3$ | 1600 | 14000 | 8000 | 2250 | 8:2 | 125 | 3 | ResNet18 |

## E.2 VISUALIZATION OF DATASETS

### E.2.1 CLASSICAL ML TASKS.

We visualize the datasets of classical ML tasks using t-Distributed Stochastic Neighbor Embedding (t-SNE)[7], as shown in Figure 5. We split ID/OOD data by setting OOD data samples as semitransparent grey dots to better observe the distinction between the ID and OOD data distributions.

### E.2.2 DL TASKS.

We also visualize the discrepancy using MMD in DL tasks since the raw features of DL tasks are hard to visualize, like Figure 5. We calculate the MMD distance within the ID data and the MMD distance between the ID and OOD data. To extract the feature representations in DL tasks, we utilized a ResNet50 which is pre-trained on ImageNet dataset[8]. After extracting the features (the output of the penultimate layer of the pre-trained ResNet50) of ID and OOD samples, we next calculate the MMD scores, which is the same as the MMD settings in our early-stopping technique. Calculating the MMD score between these ID and OOD data samples is memory-consuming (CIFAR10 and CIFAR100 have $50,000$ training sets). To calculate the ID-ID MMD score, we randomly sampled $1,000 \times 2$ instances in ID data and calculated the MMD score. We repeated this operation $100$ times and took the average score. For calculating the ID-OOD score, we perform the same operation. The MMD scores of *CIFAR10-04*, *CIFAR10-06*, *CIFAR100-04* and *CIFAR100-06* are shown in Table 2. Note that we did not calculate the MMD scores of ID-ID and ID-OOD during each stage of the AL process, since calculating the MMD scores on large-scale data with high-dimensional feature representations is time- and memory-consuming.

Table 2: MMD scores of ID-ID and ID-OOD data in DL tasks.

| | ID-ID | ID-OOD |
|---|---|---|
| CIFAR10-04 | 0.00608 | 0.04645 |
| CIFAR10-06 | 0.00596 | 0.03565 |
| CIFAR100-04 | 0.00603 | 0.01305 |
| CIFAR100-06 | 0.00604 | 0.01659 |

---

[7] https://scikit-learn.org/stable/modules/generated/sklearn.manifold.TSNE.html

[8] https://pytorch.org/vision/main/models/generated/torchvision.models.resnet50.html

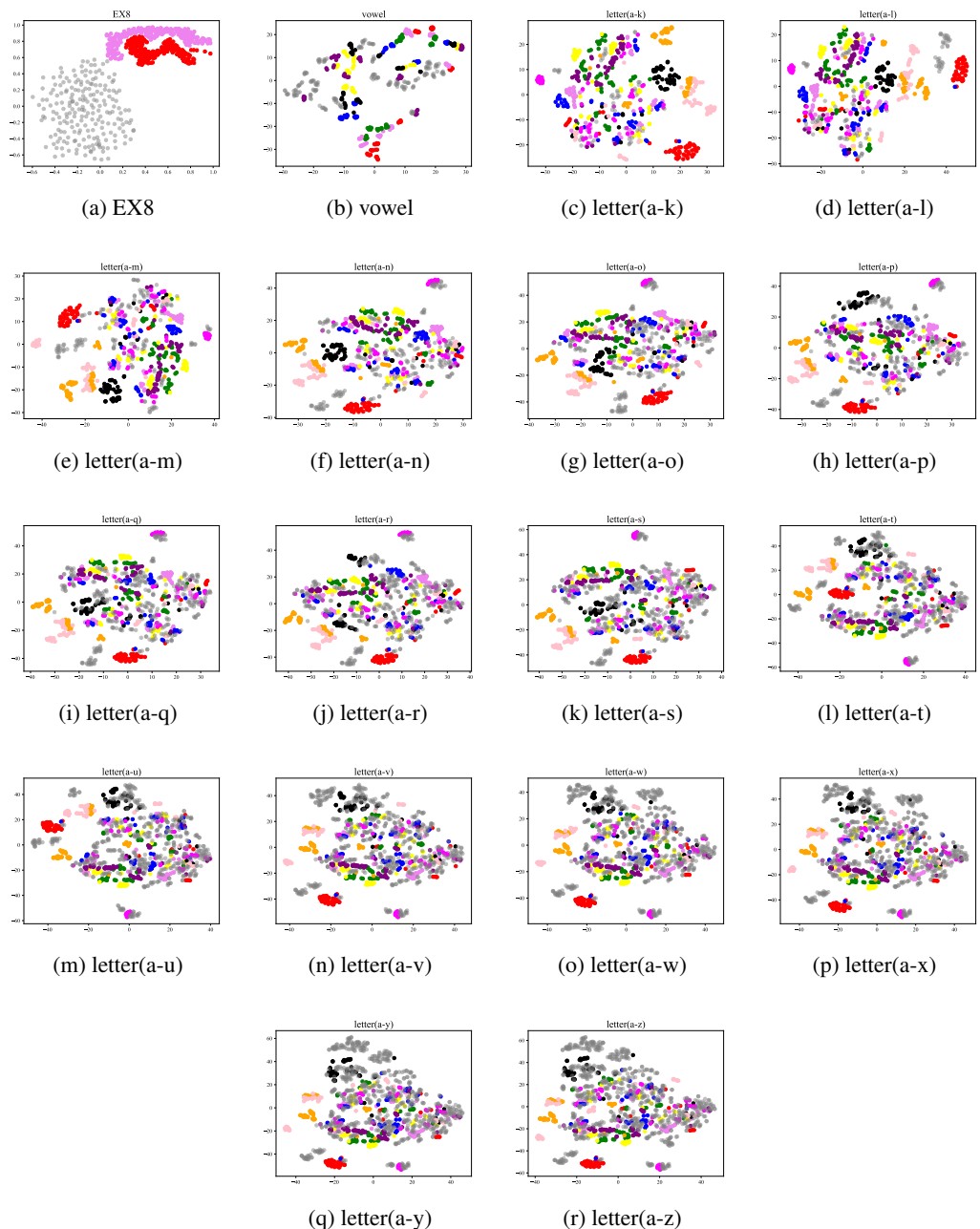

Figure 5: Visualization of data sets in classical ML tasks by t-SNE. We set the semitransparent grey dots to represent OOD data samples, the remaining colorful dots are ID data samples.

### E.3 BASELINES AND IMPLEMENTATIONS

We introduce the baselines and the implementation details in this section.

For the basic learner/classifier, we adopted Gaussian Process Classifier (GPC), Logistic Regression (LR), and ResNet18 in our experiments. We did not choose GPC in the *letter* dataset since the accuracy-budget curves based on GPC are not monotonically increasing. The requirement of selecting a basic classifier is: that the basic classifier can provide soft outputs, that is, predictive class probabilities to calculate the uncertainty of each unlabeled sample, e.g., entropy. We use the

implementation of the GPC with RBF kernel[9] and LR[10] of scikit-learn library (Pedregosa et al., 2011) with default settings. For ResNet18, we employed ResNet18 (He et al., 2016) based on PyTorch with Adam optimizer (learning rate: $1e-3$) as the basic learner in DL tasks. In *CIFAR10* and *CIFAR100* tasks, we set the number of training epochs as 30, the kernel size of the first convolutional layer in ResNet18 is $3 \times 3$ (consistent with PyTorch CIFAR implementation[11]). Input pre-processing steps include random crop (pad $= 4$), random horizontal flip ($p = 0.5$) and normalization.

For the implementation of the classical ML baselines, we have introduced it in the main paper.

For the baselines in DL tasks, we use the implementation of DeepAL+[12](Zhan et al., 2022a). We provide simple introductions of BALD, LPL and BADGE as follows:

- Bayesian Active Learning by Disagreements (BALD) (Houlsby et al., 2011; Gal et al., 2017): it chooses the data points that are expected to maximize the information gained from the model parameters, i.e., the mutual information between predictions and model posterior.
- Loss Prediction Loss (LPL) (Yoo & Kweon, 2019): it is a loss prediction strategy by attaching a small parametric module that is trained to predict the loss of unlabeled inputs concerning the target model by minimizing the loss prediction loss between predicted loss and target loss. LPL picks the top $b$ data samples with the highest predicted loss.
- Batch Active learning by Diverse Gradient Embeddings (BADGE) (Ash et al., 2020): it first measures uncertainty as the gradient magnitude for the parameters in the output layer in the first stage; it then clusters the samples by $k$-Means++ in the second stage.

For CCAL, We utilize the open-source code implementation of CCAL[13]. We train SimCLR (Chen et al., 2020), the semantic/distinctive feature extractor, which is provided by CCAL's source code. We train the two feature extractors with 700 epochs and a batch size of 32 on a single V100 GPU.

We run all our experiments on a single Tesla V100-SXM2 GPU with 16GB memory except for running SIMILAR (FLCMI) related experiments. Since SIMILAR (FLCMI) needs much memory. We run the experiments (SIMILAR on down-sampled *CIFAR10*) on another workstation with Tesla V100-SXM2 GPU with 94GB memory in total.

## F  EXPERIMENTS

This section is an extension of Section 4 in the main paper. We provide additional experimental results analysis and more experiments that do not appear in the main paper due to the page limit.

### F.1  EXPERIMENTS ON CLASSICAL ML TASKS

We present the complete accuracy vs. budget curves and numbers of OOD samples selected vs. budget curves during AL processes in Figure 6 and Figure 7 respectively. We observe from Figure 5, Figure 6, and Figure 7 that the capability of Mahalanobis distance is limited by the distinction between ID data distribution and OOD data distribution. *EX8* has distinct ID/OOD data distributions (see Figure 5-a), thus the Mahalanobis distance well distinguishes ID and OOD data, and reaches the optimal performance (Figure 7-a, MAHA has the same curve with IDEAL-ENT). However, for *vowel* and *letter* dataset, the ID/OOD data distributions are not as distinct as *EX8* (see Figure 5 b-r), thus the performance of MAHA is influenced. Furthermore, it affects the performance of our POAL. It makes our POAL behaves less perfectly on *vowel* and *letter* datasets than on the *EX8* dataset. Similar conclusions appear in (Ren et al., 2019). This is our future work to find better ways to differentiate ID and OOD data distributions.

---

[9]https://scikit-learn.org/stable/modules/generated/sklearn.gaussian_process.GaussianProcessClassifier.html
[10]https://scikit-learn.org/stable/modules/linear_model.html#logistic-regression
[11]https://github.com/kuangliu/pytorch-cifar/blob/master/models/resnet.py
[12]https://github.com/SineZHAN/deepALplus
[13]https://github.com/RUC-DWBI-ML/CCAL/

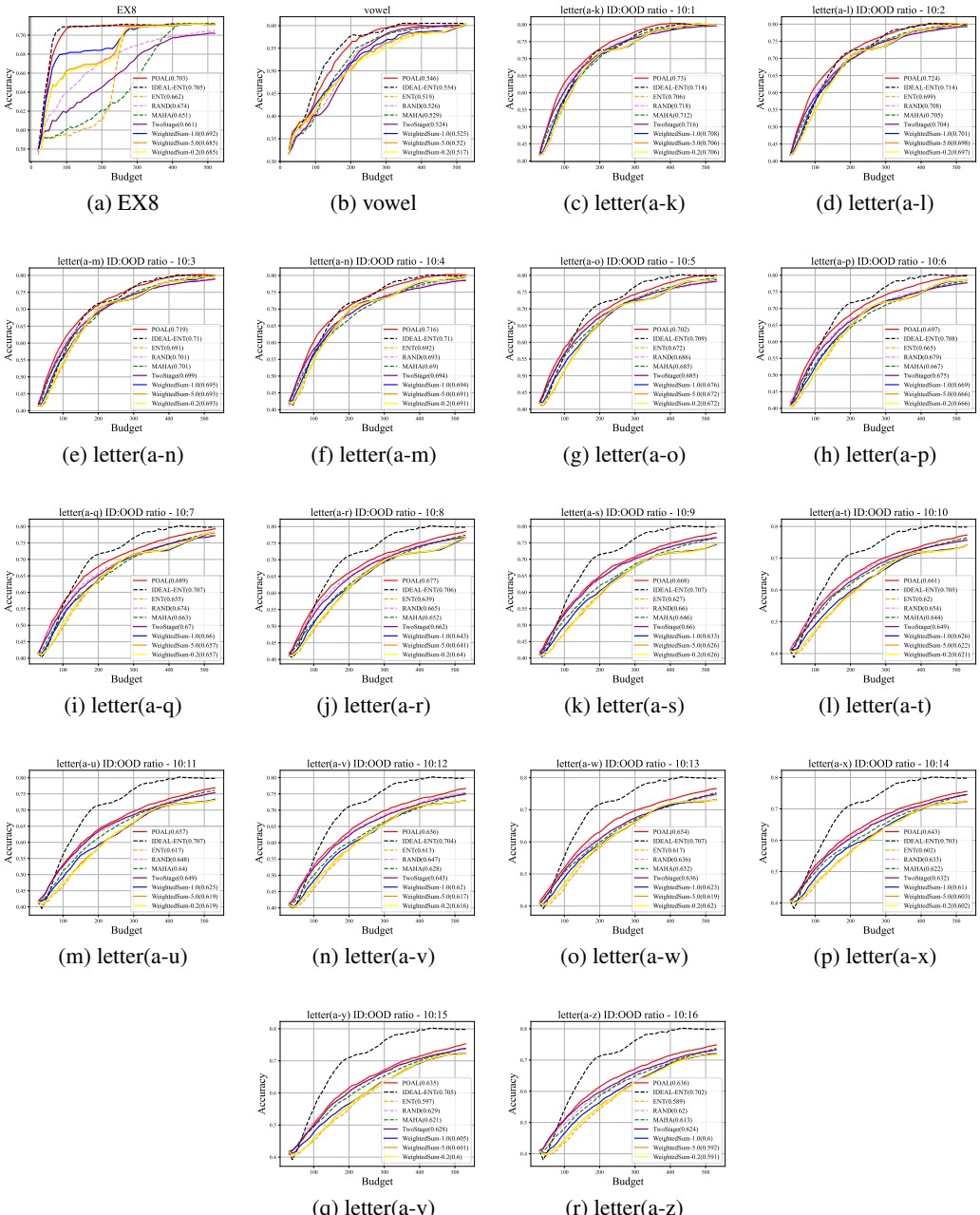

Figure 6: Accuracy vs. budget curves for classical ML tasks. The AUBC performances are shown in parentheses in the legend.

## F.2 EXPERIMENTS ON DL TASKS

The comparative results are shown in Figure 8. It is an enlarged version of Fig. 4, with the same contents. We have moderately better performance than CCAL on *CIFAR10-04* dataset, i.e., our POAL has 0.762 AUBC performance and CCAL is 0.754. Note that in Figure 8-c, POAL has similar efficiency with **CCAL** on preventing selecting OOD data samples on *CIFAR10-04* dataset. In *CIFAR10-06* task, as the OOD ratio increases, our POAL outperforms CCAL, i.e., our POAL has 0.84 AUBC performance while CCAL is only 0.819. And POAL selects less OOD samples than CCAL, as shown in Figure 8-d. On *CIFAR100* datasets, the advantages are more significant.

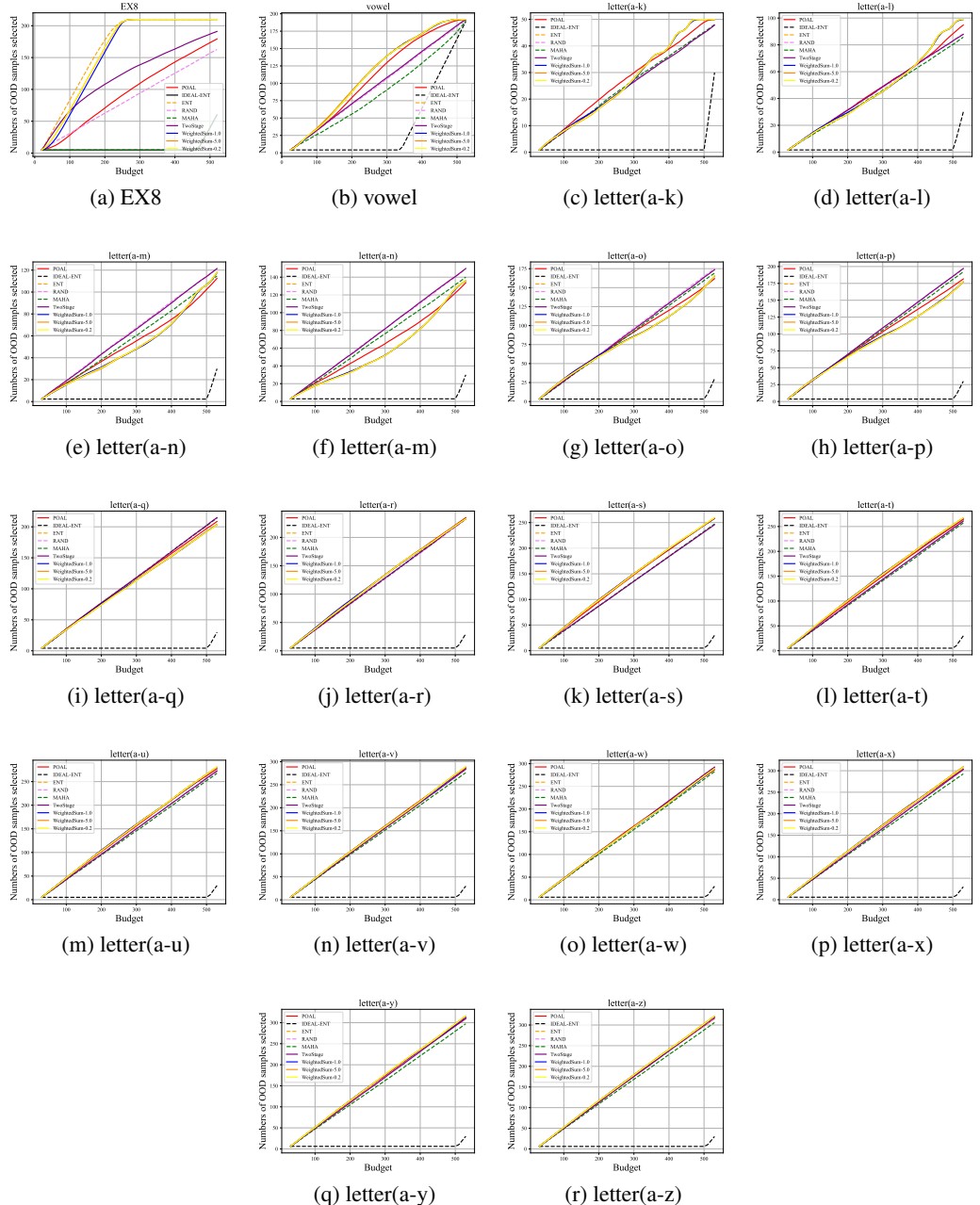

Figure 7: Numbers of OOD samples selected vs. budget curves for classical ML tasks during AL processes.

POAL selected less OOD data than CCAL. Both POAL and CCAL perform better than normal AL sampling schemes (i.e., ENT, LPL, BALD, $k$-Means, BADGE and RAND) on both *CIFAR10-04* and *CIFAR10-06* datasets.

### F.3 ADDITIONAL EXPERIMENTS (1): POAL VS. SIMILAR (FLCMI)

In the main paper, we have compared the SIMILAR with FLVMI as submodular mutual information. Referring to its comparable performance on *CIFAR10* dataset (see Figure 5a and Figure 5b in (Kothawade et al., 2021) as reference). We choose FLVMI as baselines in our main paper since it is both time and memory efficient. However, FLVMI performs worse than FLCMI, since FLCMI is

specially designed for AL under OOD data scenarios. However, FLCMI is both time and memory-consuming. In (Kothawade et al., 2021), they adopted down-sampled experiments. Although the partition trick could be applied to solve this time/memory-consuming problem, the partition will influence the final performance due to the randomness. To provide a fair comparison. We conduct additional experiments on the down-sampled CIFAR10 dataset on SIMILAR (FLCMI), following the experimental settings in (Kothawade et al., 2021).

The comparison experiments are shown in Figure 9. Besides our POAL and SIMILAR (FLCMI), we also provide more baselines as reference, i.e., IDEAL-ENT, ENT, Margin (Wang & Shang, 2014) and RAND. As shown in Figure 9, both SIMILAR (FLCMI) and POAL have better performance than normal AL sampling strategies. From the aspect AUBC evaluation metric, our model has comparable performance with SIMILAR, 0.667 vs 0.669. Nevertheless, we have a lower standard deviation value than SIMILAR, so our POAL is more stable. From the aspect of Accuracy vs. Budget curves, in early stages (e.g., Budget < 2,500), our POAL is better than SIMILAR (FLCMI) and in latter stages SIMILAR (FLCMI) exceeds POAL. The reason is that in SIMILAR (FLCMI), they calculate the similarities between the ID labeled set and the unlabeled pool and the dissimilarities between the OOD labeled set and the unlabeled pool. In the early stages, the OOD data is insufficient. Thus SIMILAR(FLCMI) will select more OOD samples, as shown in Figure 9-b. From this experiment, we find that SIMILAR (FLCMI) only performs well when we have enough information on both ID and OOD data samples, which results in more OOD data sample selection. Our POAL only considers the distance between unlabeled samples and labeled ID samples, so we are more efficient in preventing OOD sample selection. Additionally, our POAL is more widely adopted since we could be adopted on large-scale datasets. However, SIMILAR is limited by the computation condition. Our method is also more time efficient than SIMILAR (FLCMI), as shown in Table 3. SIMILAR (FLCMI) runs five times longer than our POAL.

Table 3: The mean and standard deviation of running time (in seconds) of the comparative experiments with 3 repeated trials between our POAL and SIMILAR.

| Method | POAL-PS | SIMILAR | IDEAL-ENT | ENT | Random | Margin |
|--------|---------|---------|-----------|-----|--------|--------|
| Time | 6419.0 (109.9) | 32837.7 (897.7) | 2225.0 (12.6) | 1690.0 (15.3) | 1507.7 (5.8) | 1573.7 (9.8) |

### F.4 Additional Experiments (2): Sensitivity Analysis of Hyper-parameters

In our main paper and Algorithm 1, we have a hyper-parameter $s_w$ (sliding window size) to control the early stopping condition. Large $s_w$ refers to a more strict early stop condition and vice versa. If there is no significant change within $s_w \times p_{inv}$ iterations/populations, then we can stop early. In our experiments, we set $s_w = 20$. In this section, we conduct an ablation study to see if less strict or more strict conditions influence the final performance, as shown in Table 4. We found that $s_w = 10$ (less strict) does not affect the model performance (there is no significant difference of AUBC performance between $s_w = 20$ and $s_w = 10$). Less strict early stopping also requires less running time, e.g., in *CIFAR100-04* dataset, we have 66,880.7 seconds average running time with $s_w = 20$ and 4,932.3 seconds average running time with $s_w = 10$.

Besides $s_w$, we have another hyper-parameter $s_m$, which controlls the minimum pre-selected subset size. $s_m$ is designed to adapt to various computational resource constraints but not for controlling the AL/OOD trade-off. Larger $s_m$ can be used when more computational resources/time are available, while smaller $s_m$ can save computational cost but may lead to slightly sub-optimal solutions. In our experiments, we set $s_m = 6b$ based on our computational resource situation, which works well in practice.

Although the numbers of hyper-parameters of POAL are similar to the weighted sum, their purposes are fundamentally different. In the weighted sum optimization, hyper-parameters relate to the weighting of the two objectives, significantly influencing the result. In contrast, the two hyper-parameters of POAL are about the efficient approximation (concerning computational resources), which mainly affects the initialization and stopping criteria, possibly resulting in sub-optimal approximate solutions. Note that we do not need to tune these parameters for various tasks.

Table 4: The hyper-parameter sensitivity of sliding window size $s_w$, including mean and standard deviation (SD) of AUBC performance and running time.

| CIFAR10-04 | AUBC | Time (seconds) |
|---|---|---|
| **POAL** with $s_w = 20$ | 0.7620 (0.0033) | 62082.7 (7371.9) |
| **POAL** with $s_w = 10$ | 0.7613 (0.0024) | 56107.0 (3364.2) |
| CIFAR10-06 | AUBC | Time |
| **POAL** with $s_w = 20$ | 0.8400 (0.0029) | 34946.0 (4860.2) |
| **POAL** with $s_w = 10$ | 0.8403 (0.0029) | 47362.0 (188.7) |
| CIFAR100-04 | AUBC | Time |
| **POAL** with $s_w = 20$ | 0.4807 (0.0009) | 66880.7 (1583.5) |
| **POAL** with $s_w = 10$ | 0.4797 (0.0017) | 4932.3 (1891.2) |
| CIFAR100-06 | AUBC | Time |
| **POAL** with $s_w = 20$ | 0.5253 (0.0005) | 62762.0 (4277.4) |
| **POAL** with $s_w = 10$ | 0.5270 (0.0008) | 55955.3 (5162.4) |

## F.5 ADDITIONAL EXPERIMENTS (3): MORE MULTI-OBJECTIVE OPTIMIZATION STRATEGIES

In the main paper, we have introduced weighted-sum optimization, widely adopted in multiple-criteria/objective optimization problems. There is another available similar method called Weighted-Product Model (WPM). WPM is a popular multi-criteria decision-making (MCDM) method, similar to the weighted-sum optimization model. The main difference between weighted-sum and weighted-product is that instead of adding in the main mathematical operation, it uses multiplication. We wrote WPM in AL with OOD data scenario as as $\alpha_{\textbf{WeightProd}} = \arg\max_{\textbf{s}} = \mathcal{U}(s)^{\eta} \times \mathcal{M}(s)^{(1-\eta)}$. We conduct single experiments to show the performance on classical ML tasks (i.e., *EX8* dataset). There is no significant difference between weighted-sum and weighted-product optimization (Gupta, 2022) (especially on subset selection, since the ranking of data samples are similarly produced by weighted-sum and weighted-product). The experimental results are shown in Table 5. The results using weighted product optimization is similar to weighted-sum optimization.

Table 5: The comparison between weighted-sum and weighed-product optimization method. We test the performance on *EX8* dataset.

| Method | AUBC (acc) |
|---|---|
| WeightedSum - $\eta = 0.5$ | 0.692 |
| WeightedSum - $\eta = 0.2$ | 0.685 |
| WeightedSum - $\eta = 0.8$ | 0.685 |
| WeightedProd - $\eta = 0.5$ | 0.688 |
| WeightedProd - $\eta = 0.2$ | 0.688 |
| WeightedProd - $\eta = 0.8$ | 0.691 |

## F.6 ADDITIONAL EXPERIMENTS (4): POAL INCORPORATED WITH OTHER AL METHODS

To present the flexibility of our POAL, e.g., is able to incorporate more AL sampling schemes, we conduct a simple experiment on classical ML tasks, we incorporate QBC (Seung et al., 1992) and LAL (Konyushkova et al., 2017), these two methods come from high-cited AL-related publications. The selection of basic AL sampling strategies is according to the comparative survey (Zhan et al., 2021b), in which both QBC and LAL show competitive performance across multiple tasks and AL sampling strategies. We repeat the trials ten times in each experiment. The results are presented in Figure 10. We conduct experiments on *EX8* and *vowel* datasets. This experiment shows that our POAL is flexible to handle various AL sampling strategies.

## F.7 SUMMARISING EXPERIMENTS RESULTS ON DL TASKS

To observe the efficiency of each AL sampling strategy in our experiments, we summarized the overall performance across all models and datasets we adopted, including the mean and standard deviation (SD) of AUBC performance and running time. We record the running time from the start

Table 6: Overall comparison across all AL sampling strategies of all DL tasks, including mean and standard deviation (SD) of AUBC performance and running time across three repeated trials.

| | CIFAR10-04 | | CIFAR10-06 | |
|---|---|---|---|---|
| | AUBC | Time (seconds) | AUBC | Time (seconds) |
| **RAND** | 0.7500 (0.0014) | 10805.3 (148.0) | 0.8080 (0.0008) | 5217.7 (348.0) |
| **IDEAL-ENT** | 0.8007 (0.0029) | 14578.0 (838.2) | 0.8737 (0.0019) | 9398.0 (696.4) |
| **POAL** ($s_w = 20$) | **0.7620 (0.0033)** | 62082.7 (7371.9) | 0.8400 (0.0029) | 34946.0 (4860.2) |
| **POAL** ($s_w = 10$) | 0.7613 (0.0024) | 56107.0 (3364.2) | **0.8403 (0.0029)** | 47362.0 (188.7) |
| **CCAL** | 0.7543 (0.0009) | 76129.3 (5853.7) | 0.8190 (0.0037) | 35715.3 (328.5) |
| **SIMILAR (FLVMI)** | 0.7397(0.0012) | 13942.7(116.2) | 0.7947(0.0037) | 8550.0(231.3) |
| **ENT** | 0.7350 (0.0029) | 10155.3 (872.3) | 0.7960 (0.0057) | 5485.7 (655.4) |
| **LPL** | 0.7510 (0.0071) | 12384.7 (1555.9) | 0.7873 (0.0103) | 3783.3 (326.9) |
| **BADGE** | 0.7437 (0.0029) | 41646.0 (12723.5) | 0.8063 (0.0037) | 17539.3 (628.0) |
| **KMeans** | 0.7440 (0.0014) | 18785.7 (2571.3) | 0.8100 (0.0022) | 8627.3 (407.2) |
| **BALD** | 0.7480 (0.0022) | 10478.3 (679.7) | 0.8060 (0.0022) | 4158.3 (431.1) |
| **TwoStage** | 0.7300 (0.0029) | 8342.3 (1059.4) | 0.7970 (0.0028) | 5524.7 (821.3) |
| **WeightedSum-1.0** | 0.7337 (0.0019) | 9489.7 (454.5) | 0.8033 (0.0033) | 2006.3 (13.8) |
| **WeightedSum-5.0** | 0.7327 (0.0029) | 11892.3 (1733.9) | 0.8060 (0.0045) | 7998.3 (81.6) |
| **WeightedSum-0.2** | 0.7340 (0.0045) | 8249.3 (2019.2) | 0.8063 (0.0012) | 5751.0 (795.3) |
| | CIFAR100-04 | | CIFAR100-06 | |
| | AUBC | Time (seconds) | AUBC | Time (seconds) |
| **RAND** | 0.4560 (0.0016) | 11563.3 (985.5) | 0.4453 (0.0026) | 11075.3 (1198.2) |
| **IDEAL-ENT** | 0.5250 (0.0008) | 17736.7 (1694.7) | 0.5707 (0.0017) | 19098.0 (1458.8) |
| **POAL** ($s_w = 20$) | **0.4807 (0.0009)** | 66880.7 (1583.5) | 0.5253 (0.0005) | 62762.0 (4277.4) |
| **POAL** ($s_w = 10$) | 0.4797 (0.0017) | 4932.3 (1891.2) | **0.5270 (0.0008)** | 55955.3 (5162.4) |
| **CCAL** | 0.4400(0.0008) | 21253.0(2171.9) | 0.4467(0.0024) | 65303.0(1676.6) |
| **SIMILAR (FLVMI)** | 0.4377(0.0026) | 20732.7(662.1) | 0.4510(0.0024) | 13137.0(396.7) |
| **ENT** | 0.4267 (0.0034) | 11202.3 (2484.7) | 0.4100 (0.0036) | 10940.0 (1318.4) |
| **LPL** | 0.4140 (0.0037) | 17797.0 (4093.8) | 0.4087 (0.0042) | 5633.0 (252.3) |
| **BADGE** | 0.4530 (0.0000) | 58877.7 (13876.3) | 0.4430 (0.0008) | 58046.3 (17296.9) |
| **KMeans** | 0.4527 (0.0005) | 34944.3 (4121.3) | 0.4413 (0.0021) | 17465.0 (3570.3) |
| **BALD** | 0.4467 (0.0040) | 14699.0 (2675.9) | 0.4313 (0.0061) | 4574.3 (140.5) |
| **TwoStage** | 0.4347 (0.0021) | 11484.3 (4360.1) | 0.4137 (0.0017) | 3234.3 (91.1) |
| **WeightedSum-1.0** | 0.4267 (0.0025) | 12722.0 (4164.8) | 0.4103 (0.0005) | 9455.3 (468.5) |
| **WeightedSum-5.0** | 0.4287 (0.0039) | 14822.3 (6249.7) | 0.4073 (0.0050) | 11309.3 (1303.4) |
| **WeightedSum-0.2** | 0.4290 (0.0022) | 11736.7 (4394.3) | 0.4097 (0.0033) | 8865.7 (1230.1) |

of the AL process to the output of the final basic learner. From Table 6, we can observe that our model outperforms all baselines (except for the ideal model – IDEAL-ENT) in terms of AUBC performance. For running time, compared with normal AL sampling strategies, both POAL and CCAL are incomparable. Among POAL, CCAL and SIMILAR, which are designed specific to AL under OOD data scenarios, the running cost of our POAL is affordable. Especially, as mentioned in the previous section of hyper-parameter sensitivity, the running timing cost of our POAL can be reduced by loosing the early stop conditions (less $s_w$).

## F.8 VISUALIZATION OF THE SUBSET SELECTION OF POAL

A direct way to validate the efficiency of POAL is to measure the difference between the selected subset by POAL and truly Pareto set. However, on normal datasets, the searching space of the optimal subset selection is too large, as analysed in Section 3.4, it is a combination-level $- \mathcal{C}(M, b)$. Therefore, we use the subset selected by typical Pareto optimization (the operation is the same as the operation of the first round of pre-section technique in Section 3.4) to represent the optimal subset selection instead. We visualize the subset selection on data and multi-objective space, as shown in Figure 11. Using typical Pareto optimization would i) select a subset with non-fixed subset size and ii) possibly include many OOD data samples since the entropy score of OOD data samples are relatively large.

POAL select more informative ID data less OOD samples based on current basic learners in Figure 11.

We conduct a toy experiments to show whether the subset selection generated by POAL is similar to the output generated Truly Pareto Optimization for subset selection. We down-sampled the size unlabeled data pool to 50 and set the batch size as 5, the searching space is $\mathcal{C}(50, 5) = 2, 118, 760$, which is acceptable. We transverse the whole searching space and use the same Final-selection method as in POAL to generate a single final solution. As shown in Figure 12-g, We could observe that the candidate Pareto set $\mathcal{P}$ generated by POAL is close to the selection of Truly Pareto Optimization. In Figure 12-g, the solutions generated by POAL are located on Pareto Curve (or very close to Pareto Curve), in which Pareto Curve is generated by the truly pareto optimization. Observing Figure 12a-f, the subset selection generate by POAL is better. This is because Truly Pareto Optimization finally selects the subset with higher entropy and lower ID score, thus might select OOD samples.

## G  DISCUSSIONS OF POAL

### G.1  DISCUSSIONS ABOUT HOW POAL WORKS

In our work, choosing Mahalanobis distance-based measure to calculate ID confidence score has many reasons. Besides the superiority of OOD detection performance as introduced in (Ren et al., 2019), similar to combined strategies in AL, which has been introduced in Section 2 in the main paper, we combine uncertainty-based and distance-based criteria together. This allows us to analyze the data in different aspects: task-agnostic aspect, refers to the uncertainty-based measure (e.g., **ENT**) and geometric aspect, refers to the relative distance / pair-wise similarity of unlabeled data samples. This guarantees a comparable performance under various OOD data scenarios.

We have demonstrated the effectiveness of POAL on various ratio of OOD data scenarios, as shown in Figure 6c-r. We have also conduct simple experiment on clean unlabeled data pool (no OOD data), POAL still works. We conduct a simple experiment on the *vowel* dataset, where we use all training data with 11 classes as ID data and there is no OOD data. (Note that in the AL+OOD setting, only 7 ID classes are used). On this pure ID data, the AUBC (acc) of ENT is $0.454$, and our POAL is $0.465$. Compared with normal AL, our POAL still maintains good performance when there are no OOD samples in the unlabeled data pool. This is because even if there are no OOD data in the unlabeled pool, POAL still be able to detect some outliers based on Mahalanobis distance and prevent selecting them, thus making better subset selection for AL (avoiding overfitting on outliers).

Our POAL is easily to be extended to any number of score functions, by extending the dominance relationship in Section 3.4 in the main paper, and the vector scores for the early stopping criteria. Thus, it is possible to adopt multiple AL or ID scores.

### G.2  LIMITATIONS

Currently, our POAL has several limitations: (1) The accuracy of Mahalanobis distance-based ID confidence score calculation; and (2) the Monte-Carlo sampling scheme in our POAL for seeking non-dominated fixed-sized subset solutions is not high-efficiency. In future work, we will try more suitable feature representations to construct more distinct ID-/OOD- data distributions and more convenient features like semantic and distinctive feature representations in (Du et al., 2021). Regarding the efficiency of the Monte-Carlo sampling scheme, although many sampling schemes like (adaptive) importance sampling and metropolis-hastings sampling would be more efficient than Monte-Carlo sampling, these methods might suffer from initialization problems. Since the newly generated possible candidate solutions would rely on previous selections (e.g., $\mathcal{P}$), which easily fall into local minima. Monte-Carlo sampling has no such problem. In future work, we will try to propose more efficient sampling methods for finding non-dominated subset solutions.

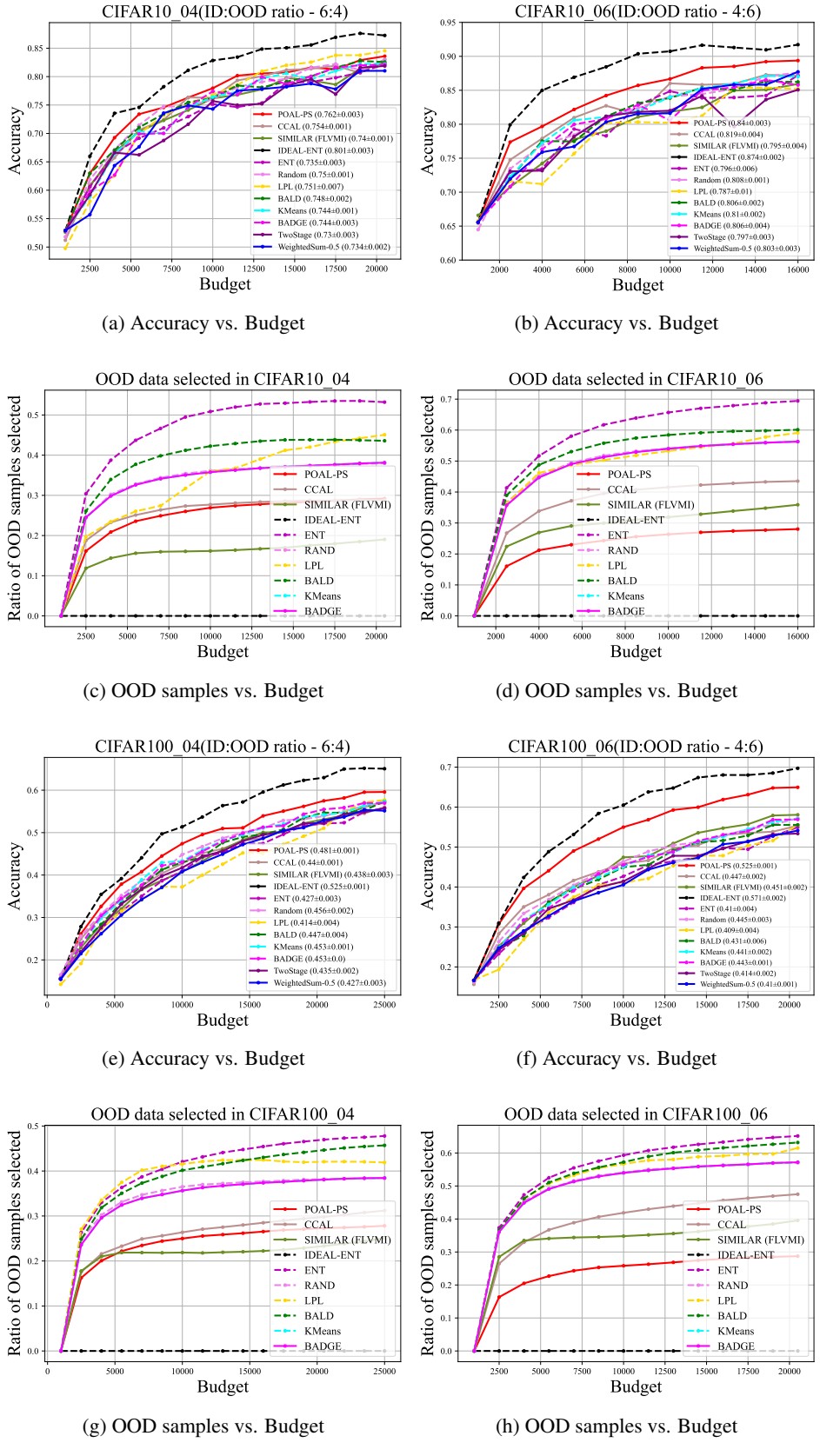

Figure 8: The comparative experiments between our **POAL-PS** and baseline methods on *CIFAR10* and *CIFAR100* datasets. This is an enlarged version of Fig. 4 in the main paper.

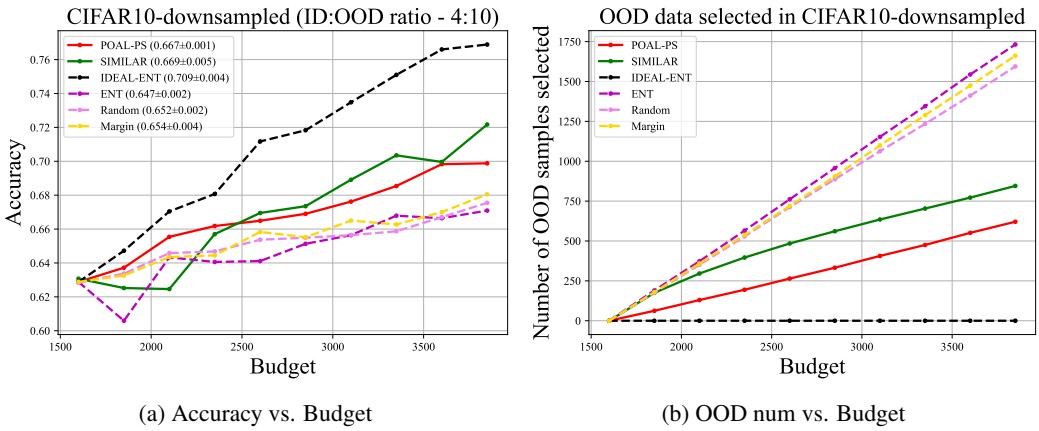

(a) Accuracy vs. Budget      (b) OOD num vs. Budget

Figure 9: The comparative experiments between our model and **SIMILAR** on down-sampled *CIFAR10* dataset.

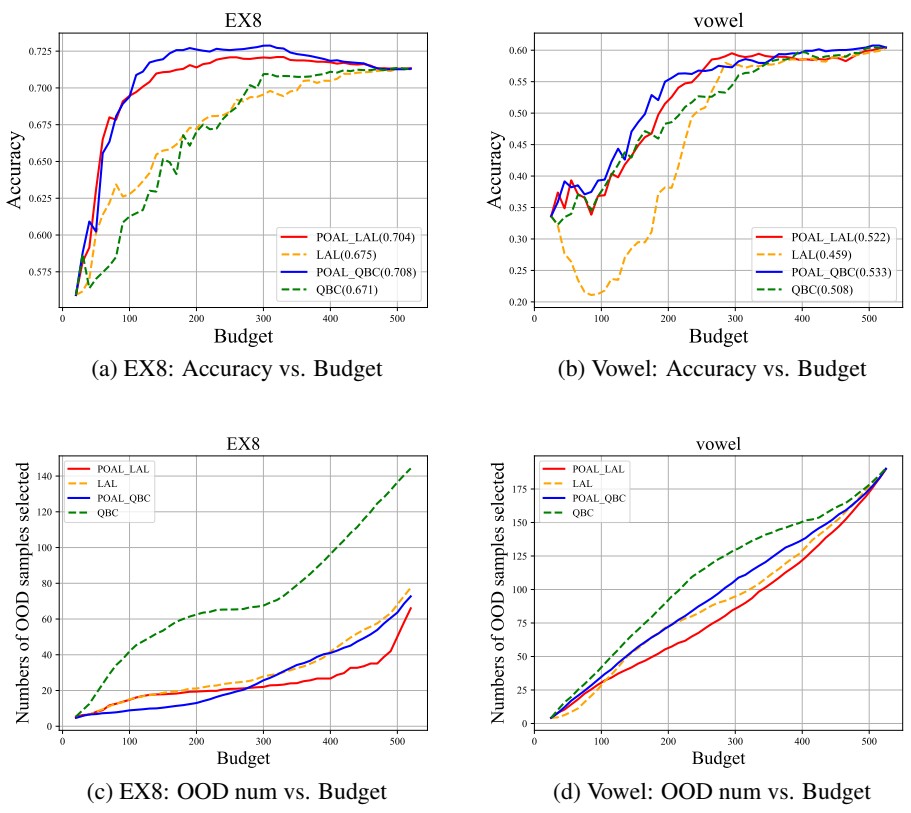

(a) EX8: Accuracy vs. Budget      (b) Vowel: Accuracy vs. Budget

(c) EX8: OOD num vs. Budget      (d) Vowel: OOD num vs. Budget

Figure 10: Flexibility: POAL incorporates various AL sampling strategies, including LAL and QBC.

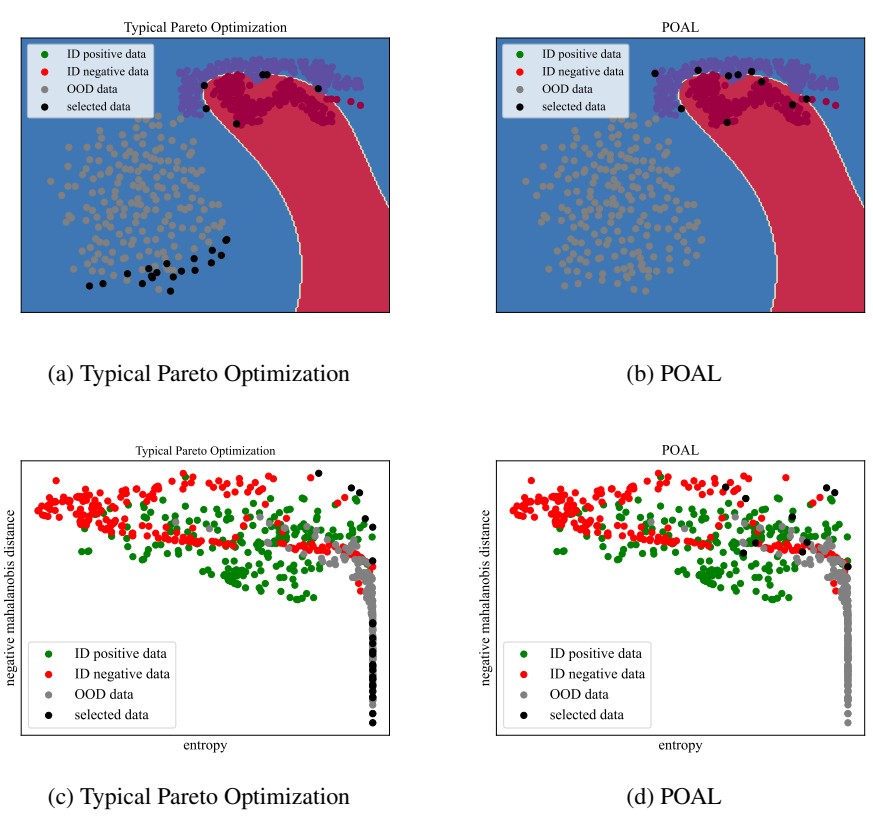

Figure 11: The subset selection generated by POAL and Typical Pareto Optimization. The selection results are presented on data space and multi-objective space respectively. The experimental settings are the same as Figure 1 in the main paper.

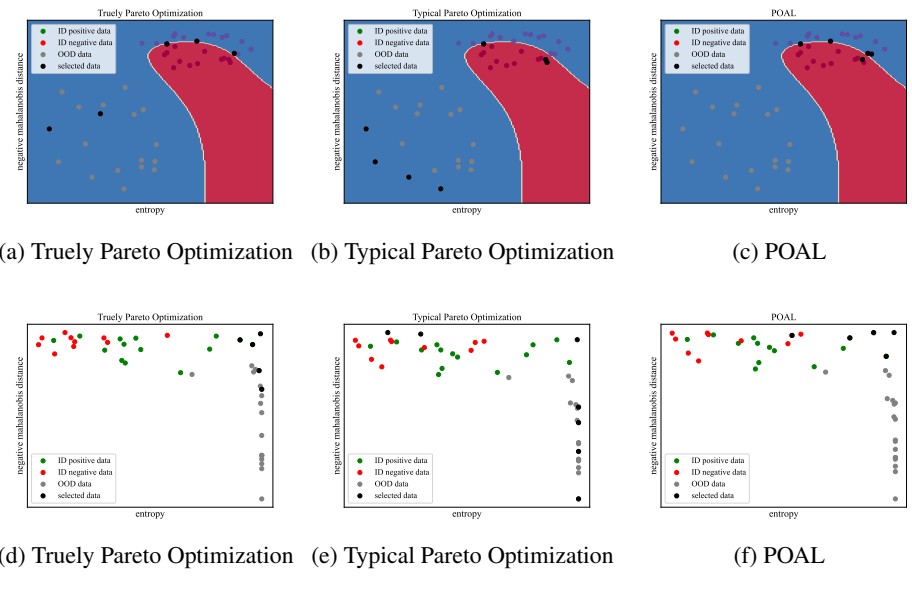

(a) Truely Pareto Optimization   (b) Typical Pareto Optimization   (c) POAL

(d) Truely Pareto Optimization   (e) Typical Pareto Optimization   (f) POAL

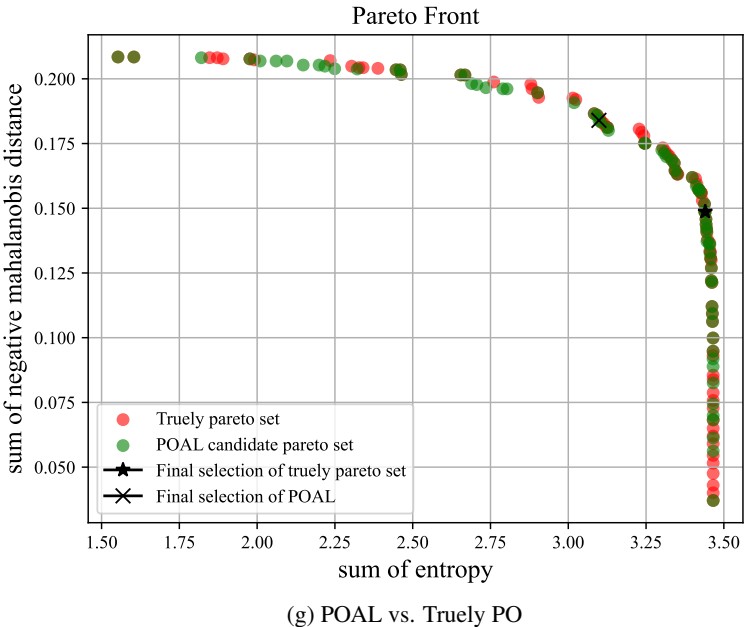

(g) POAL vs. Truely PO

Figure 12: The subset selection generated by POAL, Truely Pareto Optimzation and Typical Pareto Optimization on down-sampled *EX8* dataset. The selection results are presented on data space and multi-objective space respectively. Figure a-f are based on single data point level, Figure-g is based on subset level, where each element of pareto set is a selected subset with fixed batch size. The x-axis and y-axis are $\mathcal{U}$ and $\mathcal{M}$ respectively.

