# OpenReview forum: "Pareto Optimization for Active Learning under Out-of-Distribution Data Scenarios"
_ICLR.cc/2023/Conference — Submitted to ICLR 2023_

### Official Review · Reviewer_6RQ4 · 2022-10-24

**Confidence:** 4
**Correctness:** 3
**Technical Novelty And Significance:** 2
**Empirical Novelty And Significance:** 2
**Recommendation:** 8

**Clarity, Quality, Novelty And Reproducibility:**

Clarity: The paper is written nicely and is easy to follow.

Quality & Novelty: As discussed above, sufficient novelty is contained in the proposed method.

Reproducibility lacks sometimes.

**Strength And Weaknesses:**

Strengths:
1. This paper analyses how OOD data affects the effectiveness and efficiency of the active learning methods, and points out the conflict between AL sample selection objective and OOD detection objective.
2. This paper casts the AL sample selection under OOD data scenarios as a multi-objective optimization problem, which can automatically balance the conflict between AL and OOD objectives without tuning hyper-parameters. Experiments show its superior to other optimization strategies.
3. This paper proposes an efficient Monte-Carlo Pareto optimization algorithm for fixed-size batch-mode AL, which avoids searching a non-fixed-size Pareto Front and saves computation cost. Besides, this paper additionally adopts a pre-selection technique and an early-stopping strategy for large-scale datasets to improve efficiency.
4. The proposed optimization framework is flexible, experiments of POAL incorporated with multiple AL methods on ML and DL datasets validate its effectiveness and generality.

Weaknesses:
1. The proposed Monte-Carlo Pareto optimization mechanism for select Pareto subsets is somewhat similar to the Pareto embedding mechanism in [a], the authors can make some comparisons and analysis.
2. How to deal with the selected OOD samples? Will they be recognized by the Oracle/Annotator and then be discarded, or be wrongly annotated?
3. It will be better to visualize the selected samples on one dataset or a toy example to verify that they are truly Pareto subsets.
4. How MC POAL performs on large-scale experiments?

Refs:
[a] Karlson Pfannschmidt and Eyke Hüllermeier: Learning Choice Functions via Pareto-Embeddings. KI 2020: 327-333


**Summary Of The Paper:**

This paper explores the active learning problem under OOD data scenarios and incorporates AL and OOD objectives within a multi-objective optimization framework to balance their conflict. Specifically, they propose a Monte-Carlo Pareto optimization mechanism to enable efficient Pareto optimization, which selects optimal subsets of unlabeled samples with fixed batch size from the unlabeled data pool. The proposed framework is flexible and can apply to various combinations of AL and OOD sample selection methods, extensive experiments on ML and DL tasks demonstrate its effectiveness.

**Summary Of The Review:**

Generally, I find this paper tackled an interesting problem with a multi-objective optimization framework for active learning under ODD setting. Yet, it still requires clarification and some solid empirical support before warranting acceptance of this paper.

---

### Official Review · Reviewer_SDAU · 2022-10-24

**Confidence:** 5
**Correctness:** 1
**Technical Novelty And Significance:** 3
**Empirical Novelty And Significance:** 3
**Recommendation:** 3

**Clarity, Quality, Novelty And Reproducibility:**

In general this paper is clearly written and very easy to follow. The paper studied a novel setting in active learning, the reproducibility seems fine.


**Strength And Weaknesses:**

### Pros
- This paper considered a new setting in active learning, when unlabeled data contains out-of-distribution data (or distribution shift), the uncertainty or diversity based idea could mostly capture this unreliable data, leading to a significant performance drop.
- Empirical results and ablations support the method.
- In general, this paper is clearly written.

### Cons:
- [Main concern] I do think this paper has a clear misconception of out-of-distribution data in the context of active learning, where a rethinking on the problem setting would be important.
- About related work. In fact, active learning under distribution shift has been recently discussed in a recent deep active learning framework. Surprisingly, there is even no discussion on these related works.

### Comments on cons

1.  I do think the out-of-distribution data in active learning is vaguely defined. The problem settings in the paper assume that unlabelled data could contain different distributions than labelled data (or OOD data). Thus we need to filter these query. However, this is not necessarily correct in active learning.
- (1) In real-world active learning, the label annotation by **human** could provide the information of OOD data. For example,  labeled dataset = digits, and OOD data is classification cat/dog. Well, when machine query the image of cat, **human** will return cat rather than digits information. From this perspective, there is no need to conduct the pre-selection. If the data is indeed OOD (with different semantic information), human will return ground truth label, then we could easily detect outlier.
- (2) If the OOD is defined similar semantic information (e.g, cat/dog classification in-doors and out-doors), querying these points could be beneficial or even **improve** the robustness of the prediction (see comment 2, related work section). I would think this is the problem of vague definition of OOD data, what is the formal definition in OOD data in active learning? OOD data share the same semantic information or not ? since in active learning, the data is assumed to ask **human** to query the ground truth label, there are no similar issues in the conventional out-of-distribution data generalization/detection.

2. Another important aspect is the lack of related work. As far as I know,  the distribution shift and active learning has been recently empirical and theoretically studied. For example paper [1-3] studied active learning as a distribution shift problem (labelled data distribution and unlabelled data distribution). Specifically, they assume the unlabeled data shared similar semantic information (e.g, always dog/cat but different background rather than digits). Related theories/practice are developed. I do think these are important to discuss and compare.

[1] Deep active learning: Unified and principled method for query and training. Aistat 2020

[2] Discrepancy-Based Active Learning for Domain Adaptation. ICLR 2022

[3] Low-Budget Active Learning via Wasserstein Distance: An Integer Programming Approach. ICLR 2022


**Summary Of The Paper:**

This paper studied active learning under out-of-distribution data, which could be unreliable with the existence of OOD data. This paper proposed a multi-objective loss to simultaneously control the data uncertainty and simultaneously filter the OOD data. Empirical results justified the proposed approach


**Summary Of The Review:**

This paper studied active learning under out-of-distribution data, which could be unreliable with the existence of OOD data. The setting is quite novel for me and extensive empirical evaluations are done. However, this reviewer believes that there are several fundamental misconceptions in the problem setup. Besides, important related works are not discussed/compared, making this paper fall short of the acceptance bar.

-------------------------------------
### Post rebuttal after discussions

I would like to appreciate the effortful responses by authors. I further spend some time to further re-read the manuscripts and related work. The following is my additional feedback.

**About robust active learning.** In general, this paper aims to solve robustness in the context of active learning, where unlabelled data could be unreliable. However, checking the traditional active learning papers, these issues have been theoretically or practically discussed such as paper [1-4]. I should say, at least this setting is not entirely novel in active learning.

**About theoretical contribution** This paper aims to propose a novel practical method in active learning. This part is not applicable.

**About methodology and empirical contribution** Since this paper aims to propose a novel practical method, the requirements in this part should be significantly strong to achieve the acceptance bar. Unfortunately, after rechecking the paper, I would feel this is insufficient. In terms of practice, this paper evaluated several standard and relatively simple benchmarks such as tabular data and CIFAR100. However, recent **practical** active learning papers generally should evaluate very large scale, high-dimensional and complex datasets such as ImageNET or Open Images v6 such as [5-6]. Given the current paper is purely empirical, I do think such kinds of experiments are required. As for methodology, I would think the current version consists of different known components in different domains such as Pareto Optimization, out-of-distribution detection, etc. Why choose a specific approach in these domains? Why not other well-known methods in OOD detection? For addressing these, additional ablation studies on large-scale data such as ImageNET is necessary.

Overall, since this paper aims to propose a novel practical method, however, the empirical/practical are not sufficient to achieve the acceptance bar. Without strong, sufficient and convincing empirical results in a very-large, complex and challenging dataset, it is quite hard to convince the community to adopt it practically.

I hope my additional notes could further improve the paper quality.

[1] Robust Active Learning Strategies for Model Variability

[2] Corruption Robust Active Learning. Neurips 2021

[3] Robust Interactive Learning. JMLR 2012

[4] Active Learning with Logged Data. ICML 2018

[5] Towards Robust and Reproducible Active Learning using Neural Networks CVPR 2022

[6] Batch Active Learning at Scale. NeurIPS 2021


[Please note that I am not saying CIFAR 10/100 is not useful. If the paper mainly proposes novel theories in understanding robust active learning, it is perfectly fine to merely evaluate these simple datasets as a proof of concept. In contrast, if this paper is empirical or methodological, strong empirical validations on challenging dataset is required in modern active learning.]

---

> ### Comment · Reviewer_SDAU · 2022-11-15
> **Further discussions**
>
> Dear authors,
>
> I would appreciate your responses. I have read your rebuttal and I have the following additional remarks.
>
> 1. Concerning the definitions of out-of-distribution data and data distribution shift. Many papers in the review refer to anomalies/outliers/unrelated samples as out-of-distribution data (particularly in OOD detection papers). However, I would like to say that the usage of out-of-distribution terminology in active learning should be cautious. The initial motivation for detecting OOD samples is a lack of human supervision. In other words, a deep neural network could predict unrelated samples incorrectly but confidently **without human intervention**, which could be dangerous in many real-world scenarios. Human interactions, on the other hand, should always be considered in active learning (this is the key difference between active learning and conventional learning). Human feedback on outliers is not a bad thing from this perspective, because this information will also be returned to the machine learning system to improve the selection procedure. **Thus, when an ML system receives such outlier information, a natural idea in active learning would be to use this information to adaptively update the selection strategy to filter similar samples rather than always conducting constant pre-selection.** This is the NATURE of interactive learning.
>
> 2. Using a constant strategy to pre-select (such as ODIN or others) is not robust to distribution shift samples. In my previous review, I concentrated on two kinds of OOD samples with distribution shifts. (a) An outlier with completely different semantic information (for example, digits vs. dog), (b) similar semantic information but still different distributions (such as OOD generalization samples). I believe the current approach will likely filter type (b) samples, despite the fact that they are quite important and should be queried. Because training the same semantic information with different backgrounds could improve the model's robustness. From this viewpoint, I continue to believe that a dynamic detection strategy would be more appropriate. We never know what kind of distribution shifts will occur in an unlabeled dataset, so a dynamic strategy of filtering outliers and improving accuracy by learning OOD samples with the same semantic information may be more reasonable.

---

### Official Review · Reviewer_hK87 · 2022-10-29

**Confidence:** 3
**Correctness:** 3
**Technical Novelty And Significance:** 2
**Empirical Novelty And Significance:** 2
**Recommendation:** 6

**Clarity, Quality, Novelty And Reproducibility:**

Overall, the proposed method POAL is well-motivated and the paper is written in a clear manner that is easy to understand.
The proposed approach is moderately novel, but it has been shown to lead to consistent improvement over other existing AL schemes under various OOD scenarios.
While the performance improvement is generally modest, the evaluation results in the paper show that POAL may significantly enhance the AL performance in some cases.


**Strength And Weaknesses:**

Considering that the goal of AL to select informative unlabeled data points for labeling and the goal of OOD data point detection may conflict with each other, POAL takes a multi-objective optimization scheme to identify and select data points that are located on the Pareto frontier such that they jointly maximize the AL score (for being informative) and the confidence score for being an ID data point (hence less likely to be OOD data points).

In this work, the authors take the maximum entropy (ENT) approach for assessing the informativeness of a data point and use the negative Mahalanobis distance to measure ID confidence.
For computationally efficient Pareto optimization and batch selection, the paper proposes Monte-Carlo POAL for fixed-size batch selection, where Monte-Carlo sampling iteratively generates a candidate solutions at random and compares them against the current Pareto set to efficiently update the set of Pareto optimal data points.

The performance of POAL has been evaluated for a traditional ML model as well as a deep learning (DL) model, where performance evaluation results show somewhat modest yet consistent improvement over a number of existing AL schemes.


1.  As a performance bound, the authors show the AL performance that can be attained by oracle + ENT, where only ID data points are selected based on maximum entropy principles.
However, since AL selection and OOD detection conflict with each other and can confound the performance evaluation results, it would be important to show how the AL schemes are also combined with oracle OOD detection.
This would provide insights into what performance improvement may be attained by improving OOD detection and to what extent the AL performance can be enhanced by selecting a specific AL scheme.


2. While the paper proposes Mont Carlo POAL to make the Pareto optimization computationally more tractable, there are no results showing how scalable the proposed scheme is.
There should be a comprehensive evaluation of POAL in comparison with other AL schemes in terms of their scalability.


3. There should be further investigation and discussion regarding the OOD detection performance and the overall AL performance in OOD scenarios.
For example, while SIMILAR sometimes outperforms PAL in terms of OOD detection, its overall AL performance appears to fall behind POAL.
Sometimes POAL outperforms SIMILAR in terms of OOD detection, which again leads to better AL performance.
There should be ablation experiments to decouple these confounding factors arising from the conflict between AL selection vs. OOD detection, which will be informative for understanding the merits of the proposed method POAL and what factors contribute to its improved performance.



**Summary Of The Paper:**

When the unlabeled data pool contains out-of-distribution (OOD) samples, active learning (AL) becomes challenging as OOD samples may often be confused with informative in-distribution (ID) samples.
This paper aims to address this issue by proposing a novel AL scheme that can outperform existing schemes under OOD scenarios.
For this purpose, they propose a Monte-Carlo Pareto Optimization for Active Learning sampling scheme, called (POAL).
POAL aims to batch-select an effective subset of samples from the unlabeled data pool, thereby efficiently improving the model accuracy.


**Summary Of The Review:**

This paper proposes POAL, a Pareto optimization scheme for active learning under out-of-distribution data scenarios.
While the proposed idea is moderately novel and leads to modest AL performance improvement in the presence of OOD data in the unlabeled data pool, it is shown to consistently outperform a number of baseline AL schemes under OOD scenarios.
--------
Evaluation scores have been updated after reviewing the authors' response.

---

### Decision · Program_Chairs · 2023-01-20

**Decision:**

Reject

**Justification For Why Not Higher Score:**

N/A

**Justification For Why Not Lower Score:**

N/A

**Metareview: Summary, Strengths And Weaknesses:**

This paper considers the problem of active learning with distribution shift (i.e., unlabeled examples contain some out-of-distribution samples). The overall goal is to minimize the queries on OOD samples and use the available query budget to improve the accuracy of classifier on in-distribution samples. In each round, B queries are selected and any selected OOD samples are ignored. The main idea of the paper is to formulate this selection as a multi-objective optimization problem with active learning score (objective 1) and ID/OOD score (objective 2) and solve it using a Monte-Carlo approach. From the Pareto set of solutions at the time of convergence, the candidate solution which has highly overlapping unlabeled examples with other candidates is selected for querying. Experimental evaluation shows modest improvement over baselines.

Couple of reviewers were positive about the paper, but one reviewer provided critical comments about the overall paper. Based on my own reading of the paper, I tend to agree with most of the critical comments, which I summarize below.
- As multiple reviewers' pointed out there is existing work in terms of both theory and algorithms for the problem setting of active learning under distribution shift -- The problem setting is not novel. The paper should have done a better job of positioning the paper and also compare with the most relevant ones.
- The approach can be instantiated using any active learning score and ID/OOD score as part of the multi-objective formulation. The paper could have justified better about effective instantiations. Specifically, it seems that the method can benefit if the queried OOD samples are used to improve OOD detector.
- The technical novelty is somewhat limited and there is no theoretical guarantees. The final selection of B queries from the Pareto set of solution doesn't have a well-justified principle.
- Since there are no theoretical guarantees, one would expect strong empirical evaluation on large datasets to understand the robustness and effectiveness of this approach and/or comparison with most relevant active learning under distribution shift methods.

The paper is promising, but it falls short of acceptance for the above reasons. Hence, I recommend rejecting the paper. I strongly encourage the authors' to improve the paper based on the review feedback for resubmission.

**Summary Of Ac-Reviewer Meeting:**

N/A